# Can MLLMs Reason in Multimodality?
# EMMA: An Enhanced MultiModal ReAsoning Benchmark

Yunzhuo Hao [* 1]   Jiawei Gu [* 2]   Huichen Will Wang [* 3]   Linjie Li [* 4]
Zhengyuan Yang [4]   Lijuan Wang [4]   Yu Cheng [5]

## Abstract

The ability to organically reason **over** and **with** both text and images is a pillar of human intelligence, yet the ability of Multimodal Large Language Models (MLLMs) to perform such multimodal reasoning remains under-explored. Existing benchmarks often emphasize text-dominant reasoning or rely on shallow visual cues, failing to adequately assess integrated visual and textual reasoning. We introduce EMMA (Enhanced MultiModal reAsoning), a benchmark targeting organic multimodal reasoning across mathematics, physics, chemistry, and coding. EMMA tasks demand advanced cross-modal reasoning that cannot be addressed by reasoning independently in each modality, offering an enhanced test suite for MLLMs' reasoning capabilities. Our evaluation of state-of-the-art MLLMs on EMMA reveals significant limitations in handling complex multimodal and multi-step reasoning tasks, even with advanced techniques like Chain-of-Thought prompting and test-time compute scaling underperforming. These findings underscore the need for improved multimodal architectures and training paradigms to close the gap between human and model reasoning in multimodality. The project homepage can be accessed at https://emma-benchmark.github.io/.

## 1. Introduction

Multimodal reasoning is fundamental to human intelligence. For example, interior designers combine textual descriptions with mental imagery to optimize room layouts. Text-based reasoning allows us to analyze abstract concepts, while

---
[*]Equal contribution [1]University of Electronic Science and Technology of China [2]Sun Yat-sen University [3]University of Washington [4]Microsoft [5]The Chinese University of Hong Kong. Correspondence to: Yu Cheng <chengyu05@gmail.com>.

*Proceedings of the $42^{nd}$ International Conference on Machine Learning*, Vancouver, Canada. PMLR 267, 2025. Copyright 2025 by the author(s).

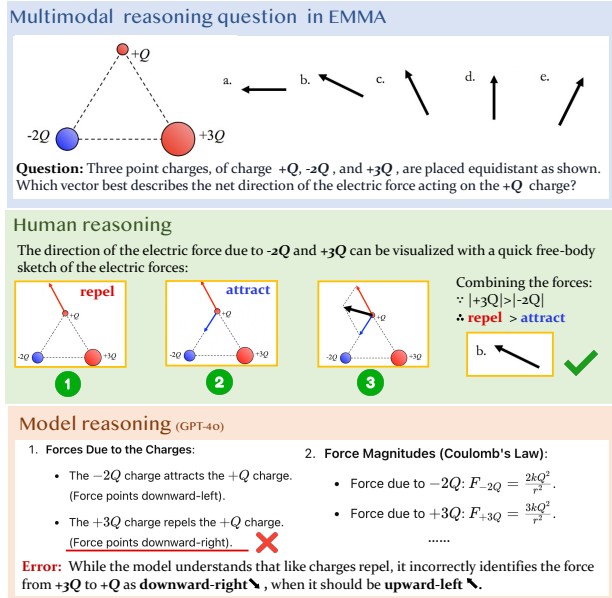

**Figure 1: A sample multimodal reasoning question in EMMA.** Humans engage in graphical reasoning: guided by the principles of electric force, they draw force vectors with appropriate directions and visually compute their sum. While GPT-4o understands that like charges repel, it mistakes the direction of the repulsive force, highlighting its limitations in multimodal reasoning.

visual reasoning enables us to draw insights from complex visual information. Combining these skills provides a robust framework for solving technical and creative challenges.

Recent advancements in Large Language Models (LLMs) have significantly enhanced their reasoning abilities (OpenAI, b; Qwen, 2024; Zhao et al., 2024; DeepSeek, 2024), enabling strong performance on tasks such as formal logic reasoning (Hendrycks et al., 2020b), graduate-level academic question answering (Rein et al., 2023), and competitive programming (codeforces; Bai et al., 2023; Hendrycks et al., 2020a). Despite these successes, these models primarily focus on *text-only reasoning*, leaving an open question: can Multimodal LLMs (MLLMs) effectively reason across both language and visual inputs?

A major bottleneck in addressing this question is the lack of appropriate benchmarks. Existing multimodal benchmarks

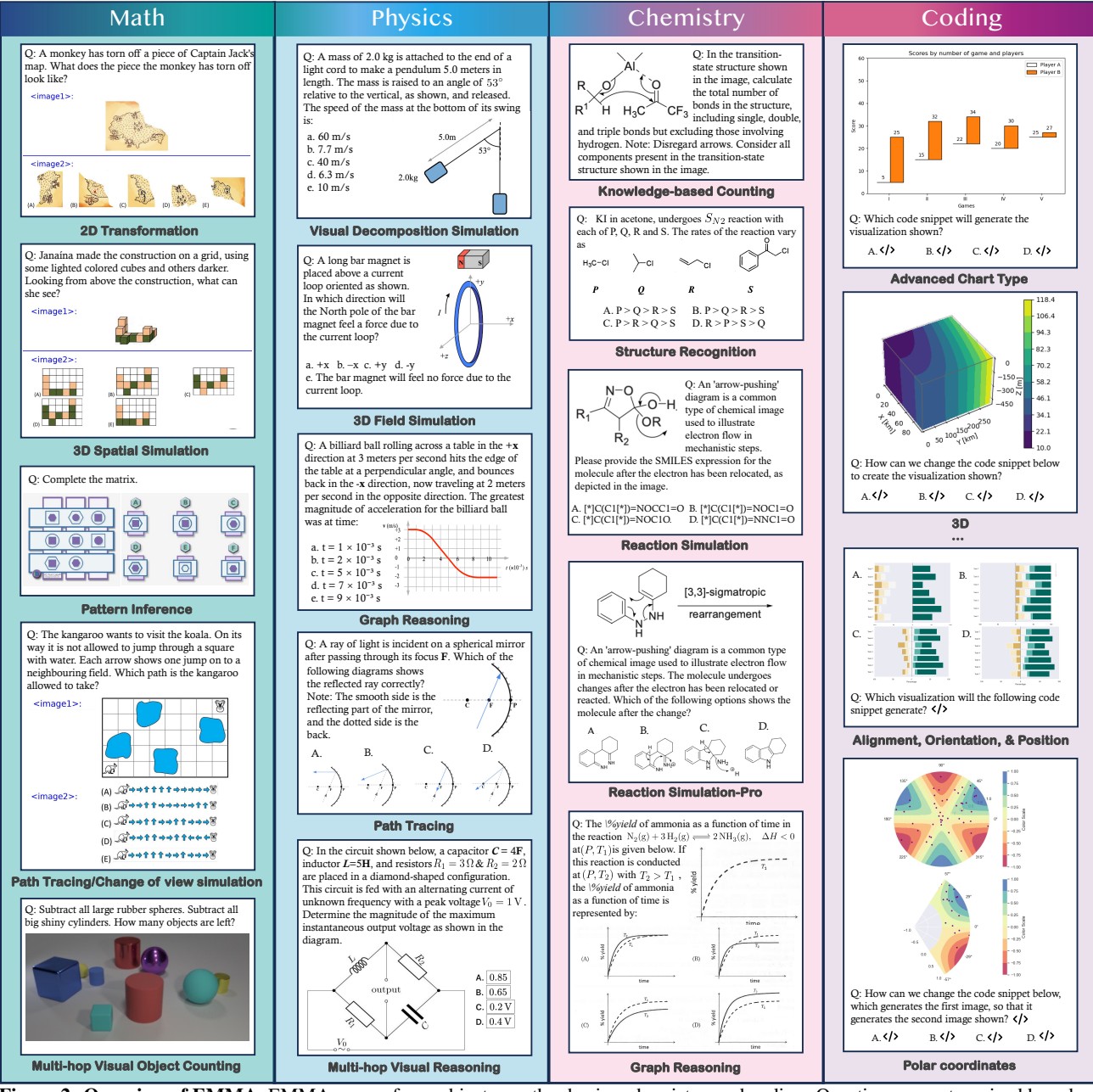

**Figure 2: Overview of EMMA.** EMMA covers four subjects: math, physics, chemistry, and coding. Questions are categorized based on the specific skills they measure, enabling fine-grained analysis.

largely test surface-level visual understanding (Antol et al., 2015; Goyal et al., 2017; Yu et al., 2023; Akter et al., 2024) or textual knowledge (Yue et al., 2024b;a; Marino et al., 2019) recall with multimodal inputs. While some benchmarks (Lu et al., 2024b; Wang et al., 2024a) include math questions with images, Zhang et al. (2024a) have shown that many of these tasks reduce to language-only reasoning, as the visual content is often fully described in text.

To address this gap, we introduce EMMA: an Enhanced MultiModal reAsoning benchmark, specifically designed to evaluate the ability to solve problems that require both visual- and language-based problem-solving. EMMA features questions that *are difficult to solve by relying solely on text-based reasoning or a single visual pass*. Instead, solving these problems necessitates a back-and-forth process between interpreting visual inputs and applying multimodal reasoning steps, where visual aids are often integral or more efficient for arriving at the solution. For instance, Figure 1 illustrates a sample physics problem that asks for the direction of the net electric force. While GPT-4o understands that like charges repel, it mistakes the direction of the repulsive force, highlighting its limitations in multimodal reasoning.

Unlike recent benchmarks (Ramakrishnan et al., 2024; Chollet, 2019), which focus on spatial cognition, or visual puzzles that can be perfectly represented in text, EMMA introduces domain-specific challenges where reasoning can often be strengthened by visual aids. These include tasks like 3D spatial transformations, chemical structure recognition, multi-step physical simulations, and program output visualization (Figure 2). EMMA consists of 992 multimodal reasoning questions gathered from existing benchmarks through a rigorous filtering pipeline, and 1,796 newly constructed questions created manually in collaboration with domain experts. Our evaluation of ten state-of-the-art (SoTA) MLLMs on EMMA reveals three key findings:

**MLLMs struggle with multimodal reasoning**: All models perform suboptimally on EMMA, regardless of the usage of Chain-of-Thought (CoT) prompting (Wei et al., 2022). On the balanced subset of EMMA, the best-performing model, Gemini 2.0 Flash Thinking, scores only 48.00%, which is 10.75% higher than the best non-reasoning MLLM, Qwen2-VL (Wang et al., 2024b), but still trails human experts by 29.75%. These results suggest a limitation of current MLLMs in performing in-depth multimodal reasoning.

**Test-time compute scaling methods with textual CoTs are insufficient**: We explore test-time compute scaling of SoTA MLLMs with different methods (e.g., majority voting, best-of-N, and tournament) up to 16 times, yet they still fail to address the multimodal reasoning challenges in EMMA. Simply increasing the number of candidate responses with textual CoTs does little to compensate for the models' inability to produce valid visual reasoning steps, particularly for tasks requiring fine-grained spatial understanding or multi-step reasoning. In addition, current MLLMs and specialized reward models struggle with complex multimodal reasoning themselves, which can make their reward signals unreliable and limit the utility of test-time compute scaling.

**Visual reasoning is the bottleneck**: Through error analysis, we find that SoTA MLLMs frequently struggle with tasks requiring precise spatial simulations, multi-hop visual reasoning, and integration of visual and textual information. These shortcomings are particularly pronounced in problems where visual aids offer a simpler path to the solution. Further, textual CoT negatively impacts model performance on visual-reasoning-heavy tasks, highlighting the need for new paradigms to improve visual reasoning.

These insights suggest that the performance gap between text-based and multimodal reasoning arises from MLLMs' limited ability to perform fine-grained visual reasoning. EMMA highlights the need for new architectures and training paradigms that can better integrate and reason over diverse modalities, enabling models to leverage both visual and linguistic information more effectively.

## 2. Related Work

**Multimodal Large Language Models** Recent years have witnessed rapid progress in MLLM development. Building upon early techniques in vision-language modeling (Tan & Bansal, 2019; Lu et al., 2019; Chen et al., 2020; Radford et al., 2021; Li et al., 2020; Zhang et al., 2021; Yu et al., 2022), modern MLLMs (Li et al., 2024a; Lu et al., 2024a; Team, c; Liu et al., 2024a; Yang et al., 2024a; Achiam et al., 2023; Team, a; Li et al., 2023) leverage the success of LLMs and various visual instruction tuning techniques (Liu et al., 2024b;a; Zhu et al., 2023), achieving impressive performance in many multimodal tasks.

**LLM and MLLM Reasoning** State-of-the-art models (OpenAI, b; Qwen, 2024; Zhao et al., 2024; DeepSeek, 2024) now achieve strong performance on tasks such as formal logic reasoning (Hendrycks et al., 2020b), graduate-level academic question answering (Rein et al., 2023), and competitive programming (codeforces). These advancements in text-based reasoning have spurred growing interest in multimodal reasoning, exemplified by visual CoT models (Shao et al., 2024; Zhang et al., 2023) and visual CoT prompting techniques (Zhou et al., 2024). Although visual CoT prompting techniques have shown promise, their focus is primarily on enhancing perception through methods like cropping images to simulate attention. Hence, these approaches offer limited support for tasks that demand more advanced visual reasoning skills, such as visual manipulation or imagination.

**Multimodal Reasoning Benchmarks** Most existing reasoning benchmarks are text-based (e.g., (Cobbe et al., 2021; Hendrycks et al., 2021; Srivastava et al., 2022; Jin et al., 2023; Suzgun et al., 2022)), but the rising demand for multimodal evaluation has led to the development of benchmarks across diverse domains (e.g., (Lu et al., 2024b; Wang et al., 2024a; Li et al., 2024b; Yang et al., 2024c; Ying et al., 2024; Chen et al., 2024a; Li et al., 2024c; Cheng et al., 2024)). Recent efforts focus on spatial and relational reasoning (Akter et al., 2024; Ramakrishnan et al., 2024) and college-level reasoning requiring domain knowledge (Yue et al., 2024a). However, many multimodal benchmarks contain redundant text-image information, allowing models to bypass visual reasoning (Zhang et al., 2024a). To address this, MMMU-Pro (Yue et al., 2024b) incorporates a filtering pipeline to enhance multimodal evaluation. In this work, we further refine such approaches by curating a benchmark that focuses explicitly on tasks requiring strong visual reasoning. Unlike existing benchmarks, our test suite emphasizes multimodal reasoning challenges that are difficult to solve with text-based reasoning and a single visual pass.

Table 1: Key statistics of EMMA.

| Statistic | Number |
|---|---|
| Total questions | 2,788 |
| - Multiple-choice questions | 2,002 (72%) |
| - Free-form questions | 786 (28%) |
| - Questions with answers | 2,788 (100%) |
| - Questions newly added | 1,796 (64%) |
| Image in the question | 2,599 (93%) |
| Image in the option(s) | 195 (7%) |
| Problems with multiple images | 298 (10%) |

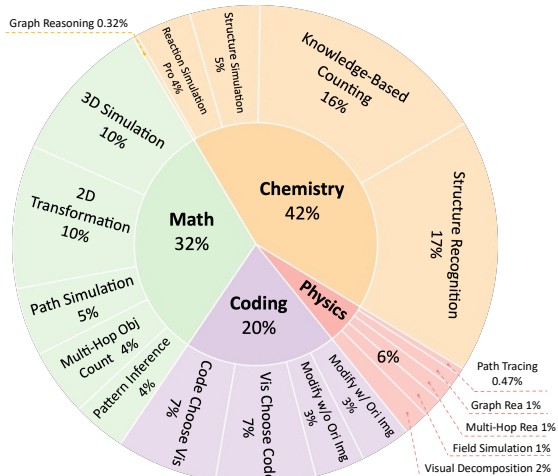

Figure 3: Composition of EMMA.

## 3. The EMMA Benchmark

### 3.1. Overview of EMMA

We introduce EMMA, an Enhanced MultiModal ReAsoning Benchmark. EMMA is composed of 2,788 problems, of which 1,796 are newly constructed, across four domains: math, physics, chemistry, and coding. All questions in EMMA are in either multiple-choice or open-ended formats with short, easily checkable ground truth answers. The key statistics of EMMA are summarized in Table 1, and its composition is presented in Figure 3.

To provide fine-grained insights into how MLLMs might fail in multimodal reasoning, we assign labels to each problem in our benchmark. These labels are either created by domain experts or assigned by GPT-4o and subsequently verified by experts. As shown in Figure 2, questions in EMMA assess a wide array of multimodal reasoning skills. For example, the pattern inference problem in math challenges models to identify and generalize visual patterns; the visual decomposition simulation problem in physics requires graphically decomposing forces to determine resultant effects; the reaction simulation problem in chemistry demands precise interpretation and simulation of electron movement; the 3D visualization problem in coding[1] evaluates spatial imagination by requiring models to associate function calls with their corresponding 3D representations.

### 3.2. Data Curation

As discussed in Section 2, most existing multimodal reasoning benchmarks likely contain many problems that primarily measure text-based reasoning. To address this, we employ a two-step approach to constructing EMMA (Figure 4). First, we source problems from existing multimodal reasoning benchmarks and apply rigorous filtering to exclude those solvable through text-based reasoning and a single visual pass. Next, we categorize the remaining problems for each subject into fine-grained multimodal reasoning skill taxonomies and manually collect more samples aligned with these taxonomies to expand our dataset.

---

[1]Different from the other subjects in EMMA, coding questions can be assigned more than one category since our visualizations tend to employ multiple advanced techniques.

**Filtering Mechanisms** To filter for questions that require multimodal reasoning, Yue et al. (2024b) provide only the text from multimodal reasoning questions to LLMs and discard questions that can be correctly answered this way. Nonetheless, some of the remaining questions may still not *truly* measure visual reasoning, as a single pass of visual perception and language understanding may suffice to answer them. We extend this one step further (illustrated in Figure 4): we first caption the images in multimodal reasoning questions using GPT-4o and then pass both the original text and our generated captions to MLLMs, filtering out questions that can be answered under this condition. For each candidate question, we query Llama-3-70B-Instruct (Dubey et al., 2024), GPT-4o, and Qwen2-72B-Instruct (Yang et al., 2024a) ten times; if any model answers a question correctly at least five times, we discard it, following the 5/10 threshold in MMMU (Yue et al., 2024a). This more stringent filtering ensures that the remaining questions require models to engage deeply with visual information. We introduce the data collection process for each project in detail below.

**Math** We first apply the filtering pipeline to MathVision (Wang et al., 2024a) and MathVista (Lu et al., 2024b), and then manually inspect the remaining set and craft a taxonomy consisting of five categories with a strong focus on multimodal reasoning, including 3D Simulation, 2D Transformation, Path Tracing, Multi-hop Object Counting, and Pattern Inference. Next, we use GPT-4o to categorize all questions based on this taxonomy, followed by a manual verification. In addition, we supplement our benchmark with additional pattern inference questions from RAVEN (Zhang et al., 2019), which inherently require multi-hop visual reasoning. This process results in a total of 892 math questions.

**Physics** We apply the filtering pipeline to multimodal physics problems in OlympiadBench (He et al., 2024), EXAMS-V (Das et al., 2024), and MMMU (Yue et al., 2024a), which yields only 80 problems. In addition, we manually collect more problems online from Learn AP

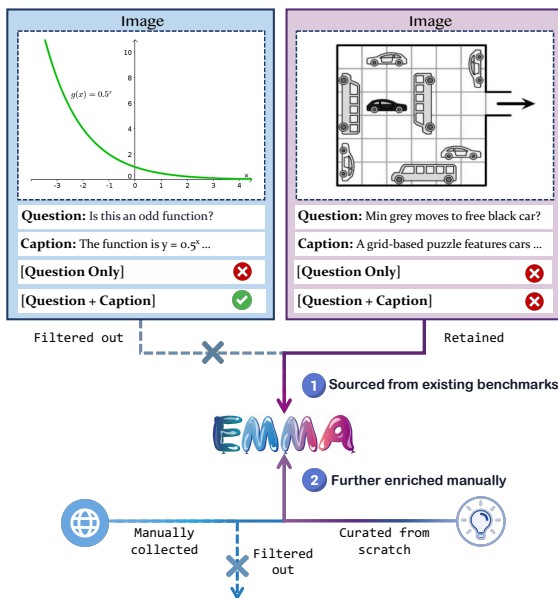

Figure 4: **Data curation process for EMMA** (Sec. 3.2).

Physics (Physics) and Khan Academy (Academy) and filter them, resulting in 76 more new problems. Through manual labeling, we verify that these problems span a wide range of topics, including 3D Field Simulation, Graph Reasoning, and Path Tracing. We note that despite our best efforts, multimodal physics problems meeting our criteria are difficult to source and construct.

**Chemistry** After filtering the chemistry portion of MMMU (Yue et al., 2024a) and EXAMS-V (Das et al., 2024), we are only left with 42 questions. Since these questions mostly involve reasoning about molecular formulas, we construct more problems on organic chemistry. We analyze the chemical properties of molecules in SMiCRM (Leung et al., 2024) with RDKit (lan, 2013), a computational chemistry toolkit, and develop 904 novel questions over chemical structure recognition and bond counting. In addition, we draw from the collection of chemical reactions in Li (2009) and collaborate with PhD students in chemistry to annotate reaction outcomes, contributing another 210 questions on reaction simulation.

**Coding** We design four coding tasks to assess MLLMs in real-world visualization creation scenarios. For instance, to evaluate the ability to reproduce a visualization, we construct "Vis Choose Code" questions where models select the code that generates a target chart. Since previous visualization design benchmarks all use MLLM judges, we do not source from them, but manually construct all coding questions from scratch. We first identify "seed visualizations" using advanced visualization techniques from CharXiv (Wang et al., 2024c), the matplotlib example gallery (Team, b) following (Wu et al., 2024), and our prior experience. Next, we generate four variations for each seed visualization to form a "set" by introducing design variations (e.g., changes in

spine configuration, line style, and axis scaling) either manually or through prompting MLLMs, with post-hoc manual verification. We provide these design variations as labels for each problem. Finally, we construct different types of questions using these visualization sets. Our curation results in 564 multiple-choice coding questions.

### 3.3. Comparison with Existing Benchmarks

Our enhanced data filtering pipeline ensures that EMMA focuses on questions requiring in-depth multimodal reasoning, i.e., those that cannot be solved solely using text-based reasoning or a single visual pass. While MMMU-Pro (Yue et al., 2024b) removes questions solvable through their text portion alone, it may still retain problems for which visual reasoning is inessential. In contrast, EMMA applies a stricter filtering criterion, discarding questions solvable with text and image captions. For instance, the left example in Figure 4 (adapted from MathVista) asks whether a depicted function is even or odd. Although unsolvable without the image, the problem can be shortcut by extracting the function's text expression embedded in the image. In this case, the role of vision is more aligned with visual perception than with visual reasoning. By eliminating such problems, which MMMU-Pro's approach would retain, EMMA better evaluates the multimodal reasoning capabilities of models.

We also contribute 1,796 novel multimodal reasoning problems across physics, chemistry, and coding. After filtering physics and chemistry problems from all relevant benchmarks to our knowledge (e.g., (He et al., 2024; Das et al., 2024; Yue et al., 2024a)), only 100 remain. We expand this to 1,332 in EMMA by manually sourcing additional data and hiring domain experts. For coding, EMMA is the first benchmark to systematically evaluate data visualization skills using a multiple-choice format, enabling a standardized assessment and obviating the need for MLLMs as judges. Moreover, through meticulous manual labeling or verification, we provide fine-grained labels for each question (Figure 2), categorizing them based on the specific skills they assess. These labels enable a detailed analysis of MLLM performance, as we demonstrate in Section 5.

## 4. Experiments

### 4.1. Evaluation Settings

**Data Split** To create a more balanced subset of EMMA, we randomly sample 400 questions (100 per subject) from the benchmark, hereafter referred to as EMMA-mini. Within each subject, we aim for equal representation across categories to the extent possible.

**Human Performance** To estimate expert-level performance on EMMA-mini, we hire two human experts per subject and report their average score. This score serves as a baseline

**Table 2: Evaluation results of state-of-the-art MLLMs, which are outperformed by human experts with wide margins.** The highest and second-highest scores are highlighted in green and blue, respectively. ↑ indicates CoT improvements, while ↓ denotes reductions.

| | CoT | EMMA | | | | | EMMA-mini | | | | |
|---|---|---|---|---|---|---|---|---|---|---|---|
| | | Math (892) | Phys. (156) | Chem. (1,176) | Coding (564) | Overall (2,788) | Math (100) | Phys. (100) | Chem. (100) | Coding (100) | Overall (400) |
| Random choice | – | 14.01 | 25.64 | 16.50 | 25.71 | 18.08 | 13.00 | 23.00 | 27.00 | 28.00 | 22.75 |
| Human Expert | – | – | – | – | – | – | 75.00 | 64.50 | 86.00 | 85.50 | 77.75 |
| Claude 3.5 Sonnet | ✗ | 25.34 | 33.97 | 40.90 | 38.65 | 35.08 | 23.00 | 34.00 | 44.00 | 35.00 | 34.00 |
| Gemini 2.0 Flash | ✗ | 23.88 | 38.46 | 36.31 | 42.02 | 33.61 | 20.00 | 40.00 | 36.00 | 41.00 | 34.25 |
| GPT-4o | ✗ | 27.24 | 38.46 | 31.89 | 40.07 | 32.42 | 30.00 | 38.00 | 33.00 | 40.00 | 35.25 |
| Qwen2-VL-72B-Instruct | ✗ | 33.07 | 42.31 | 32.06 | 34.57 | 33.46 | 38.00 | 40.00 | 34.00 | 37.00 | 37.25 |
| LLaVA-Onevision-72B | ✗ | 27.69 | 35.90 | 25.26 | 28.72 | 27.33 | 25.00 | 32.00 | 24.00 | 28.00 | 27.25 |
| InternVL2-Llama3-76B | ✗ | 25.11 | 22.44 | 24.06 | 27.84 | 25.07 | 31.00 | 22.00 | 21.00 | 28.00 | 25.50 |
| InternVL2.5-78B | ✗ | 31.39 | 38.46 | 35.20 | 31.91 | 33.50 | 30.00 | 40.00 | 38.00 | 33.00 | 35.25 |
| Claude 3.5 Sonnet | ✓ | 29.37 | 41.03 | 41.07 | 40.60 | 37.23 (↑ 2.15) | 30.00 | 38.00 | 41.00 | 39.00 | 37.00 (↑ 3.00) |
| Gemini 2.0 Flash | ✓ | 25.90 | 38.46 | 24.66 | 40.96 | 29.12 (↓ 4.48) | 24.00 | 41.00 | 36.00 | 44.00 | 36.25 (↑ 2.00) |
| GPT-4o | ✓ | 25.56 | 43.59 | 33.67 | 39.01 | 32.71 (↑ 0.29) | 27.00 | 44.00 | 35.00 | 38.00 | 36.00 (↑ 0.75) |
| Qwen2-VL-72B-Instruct | ✓ | 27.69 | 34.62 | 24.57 | 29.43 | 27.12 (↓ 6.35) | 35.00 | 34.00 | 32.00 | 23.00 | 31.00 (↓ 6.25) |
| LLaVA-Onevision-72B | ✓ | 22.42 | 15.38 | 22.70 | 30.67 | 23.82 (↓ 3.52) | 23.00 | 26.00 | 23.00 | 29.00 | 25.25 (↓ 2.00) |
| InternVL2-Llama3-76B | ✓ | 22.20 | 32.05 | 19.73 | 30.32 | 23.35 (↓ 1.72) | 27.00 | 33.00 | 21.00 | 32.00 | 28.25 (↑ 2.75) |
| InternVL2.5-78B | ✓ | 25.56 | 39.74 | 27.47 | 25.18 | 27.08 (↓ 6.42) | 31.00 | 36.00 | 24.00 | 19.00 | 27.50 (↓ 7.75) |
| Gemini 2.0 Flash Thinking-1219 | – | 31.61 | 56.41 | 37.93 | 43.44 | 38.06 | 35.00 | 57.00 | 41.00 | 41.00 | 43.50 |
| Gemini 2.0 Flash Thinking-0121 | – | 37.11 | 60.26 | 41.58 | 48.05 | 42.50 | 34.00 | 63.00 | 47.00 | 48.00 | 48.00 |
| QVQ-72B-Preview | – | – | – | – | – | – | 34.00 | 39.00 | 24.00 | 31.00 | 32.00 |
| o1 | – | – | – | – | – | – | 41.00 | 49.00 | 40.00 | 53.00 | 45.75 |

contextualizing model performance.

**Models** We evaluate ten state-of-the-art MLLMs under the zero-shot setting, including five open-source models (Qwen2-VL (72B) (Wang et al., 2024b), QVQ-72B-Preview (Alibaba), LLaVA-Onevision (72B) (Li et al., 2024a), InternVL2 (76B) (Team, c), and InternVL2.5 (78B) (Chen et al., 2024b)) and five proprietary ones (GPT-4o (OpenAI, a), Claude 3.5 Sonnet (Anthropic), Gemini 2.0 Flash (Deepmind, a), Gemini 2.0 Flash Thinking (Deepmind, b), and o1 (OpenAI, b)). Due to rate limits, we report o1 and QVQ performance on EMMA-mini only. All other models are evaluated on the entire benchmark.

**Prompting Strategies** For all models except o1, QVQ, and Gemini 2.0 Flash Thinking, we test two prompting strategies: (1) *Direct* prompting, which instructs models to output the answers without reasoning steps; and (2) Chain-of-Thought (*CoT*) prompting (Wei et al., 2022), where we prompt models to "think step-by-step".

### 4.2. Main Results

**Are MLLMs Multimodal Reasoners?** Table 2 demonstrates that all models perform suboptimally across the subjects in EMMA. On EMMA-mini, the best-performing model, Gemini 2.0 Flash Thinking-0121, achieves an accuracy of 48.00%, trailing human experts by 29.75%. At the lower end, LLaVA-OneVision-72B scores only 25.25%, barely surpassing random choice by 2.5%. Drilling down into subjects, the best models show the smallest gap with human performance on physics, with Gemini 2.0 Flash Thinking scoring 1.5% lower than human experts. This smaller gap may reflect the inherent difficulty of physics problems,

leading human experts to achieve a score of 64.5%. For other subjects, however, **the best-performing models lag significantly behind human experts, with gaps of 34%, 39%, and 32.5% in math, chemistry, and coding, respectively.** These results underscore the limitations of current MLLMs in addressing complex multimodal reasoning tasks.

On the full EMMA benchmark, closed-source models generally outperform open-source ones, particularly with CoT prompting. Across all subjects, Qwen2-VL-72B-Instruct is the only open-source model to place in the top two for any subject. Gemini 2.0 Flash Thinking-0121 scores best on every subject (note that o1 is not evaluated on the full set due to rate limits). In particular, it performs exceptionally well in physics, leading by 16.67% over the second-best model (excluding Gemini 2.0 Flash Thinking-1219), GPT-4o. Coding is another area where it excels, leading the second-best model (excluding Gemini 2.0 Flash Thinking-1219), Gemini 2.0 Flash, by 6.03%. Notably, on EMMA-mini, o1 outperforms Gemini 2.0 Flash Thinking-0121 by 5% in coding, suggesting even stronger coding capabilities. In sum, these results highlight the advantages of optimizing models for reasoning by training them to generate thought processes over traditional MLLM paradigms.

**Does CoT help?** We observe divergent tendencies in the effectiveness of CoT prompting across closed- and open-source models. We exclude o1, QVQ, and Gemini 2.0 Flash Thinking from this analysis, as they inherently generate CoT as part of their responses. Under direct prompting, accuracies achieved by the best open-source models are well within 2% of Claude 3.5 Sonnet (the best closed-source model). However, the gap widens significantly under CoT prompting, with the best open-source model underperforming by al-

**Table 3: Results of different test-time scaling strategies on EMMA-mini**, including (1) majority voting, (2) best-of-N, (3) tournament-style best-of-N. Pass@N accuracies are included as upper bounds of the scaling performance.

| Model | Method | Reward Model | N=1 | N=2 | N=4 | N=8 | N=16 |
|---|---|---|---|---|---|---|---|
| GPT-4o | Majority Voting | – | | – | 37.25 | 36.25 | 38.25 |
| | BoN | GPT-4o (Self) | | 35.50 | 35.75 | 36.75 | – |
| | BoN | Gemini Flash Thinking | 36.00 | 40.75 | 36.25 | 36.5 | – |
| | Tournament | Gemini Flash Thinking | | 40.75 | 39.25 | **41.25** | 35.25 |
| | Pass@N | – | | 45.00 | 53.25 | 65.75 | 74.00 |
| Gemini 2.0 Flash | Majority Voting | – | | – | 37.75 | 39.25 | 39.75 |
| | BoN | Gemini Flash (Self) | | 38.25 | 36.50 | 36.00 | – |
| | BoN | Gemini Flash Thinking | 36.25 | 36.75 | 37.00 | 40.25 | – |
| | Tournament | Gemini Flash Thinking | | 36.75 | 37.25 | **40.75** | 38.75 |
| | Pass@N | – | | 45.25 | 56.25 | 64.50 | 75.00 |
| Gemini 2.0 Flash Thinking | Majority Voting | – | | – | 48.00 | 49.00 | **50.75** |
| | Tournament | Gemini Flash Thinking (Self) | 43.50 | 45.50 | 47.25 | 47.25 | 48.00 |
| | Pass@N | – | | 53.75 | 64.50 | 71.50 | 81.50 |
| o1 | – | – | **45.75** | – | – | – | – |

most 10%. Comparing each model's performance with and without CoT on EMMA and EMMA-mini, **CoT prompting generally improves performance for closed-source models and reduces performance for open-source models**. Notably, Qwen2-VL-72B-Instruct and InternVL2.5-78B, the top two open-source models overall under direct prompting, suffer decreases of over 6% in accuracies on both EMMA and EMMA-mini. While some tasks might not benefit significantly from textual CoT, we hypothesize that this divergence arises because open-source models fail to fully leverage the potential of language to assist in multimodal reasoning tasks where language could be helpful. We elaborate on this hypothesis in detail in Section 5.

### 4.3. Results on Test-Time Compute Scaling

We test three test-time compute scaling methods (Snell et al., 2024; Yang et al., 2024b; OpenAI, b) on EMMA-mini: majority voting, Best-of-N selection, and Tournament selection. Both Best-of-N and Tournament selection require a reward model to select the best response among multiple candidates. We use CoT prompting to generate the candidate responses, so that the reward model has enough context to score the responses. For each test-time scaling method, we experiment with N = 1, 2, 4, 8, and 16, as long as the context length of the reward model allows.

**Majority Voting:** Majority voting selects the most frequent response among batches of N candidate responses, breaking ties by randomly choosing from the most frequent answers.

**Best-of-N:** Best-of-N selection (Cobbe et al., 2021; Lightman et al., 2023) chooses the highest-scoring response (Gu et al., 2024) according to a reward model. We explore two configurations: using the base model itself or a stronger reasoning model (e.g., Gemini 2.0 Flash Thinking) as the reward model.

**Tournament Selection:** In Tournament Selection (Son et al., 2024; OpenAI, c), responses compete in matches, with winners advancing through rounds until the final selection. We

use the best-performing reward model identified in the Best-of-N experiments, which is Gemini 2.0 Flash Thinking.

**Does test-time compute scaling help?** Table 3 presents results of test-time compute scaling methods on top of GPT-4o, Gemini 2.0 Flash, and Gemini 2.0 Flash Thinking-1219. **Overall, test-time compute scaling improves model performance, but it fails to close the gap to human expert performance.** The highest accuracy improvements are 5.25% for GPT-4o, 4.5% for Gemini 2.0 Flash, and 7.5% for Gemini 2.0 Flash Thinking. Notably, without test-time scaling (N=1), Gemini 2.0 Flash Thinking-1219's accuracy is 2.25% lower than that of o1, but it overtakes o1 by 5% with majority voting at N=16. Nonetheless, its best performance still lags human performance by 27%.

We also observe distinct patterns in test-time compute scaling performance across different models. While scaling beyond N=8 for GPT-4o and Gemini 2.0 Flash leads to performance degradation, Gemini 2.0 Flash Thinking continues to benefit incrementally from additional test-time compute, at least up to N=16. In fact, stronger base models also achieve higher Pass@N accuracy: Gemini 2.0 Flash Thinking's Pass@N consistently surpasses those of the other two models by around 7%, suggesting that a stronger base reasoner is more likely to cover the correct response when given multiple attempts. In sum, these results suggest that using a stronger model as the base model raises the upper bound for test-time scaling.

Comparing scaling strategies for each model, we find that GPT-4o and Gemini 2.0 Flash achieve their greatest improvements when Gemini 2.0 Flash Thinking is used as the reward model. Additionally, tournament selection consistently outperforms Best-of-N (BoN) selection. These results suggest that using a stronger model as the reward model enables weaker models to achieve better results, particularly when the reward model can make fine-grained decisions involving a couple candidate responses each time. This is intuitive, as evaluating responses also requires reasoning.

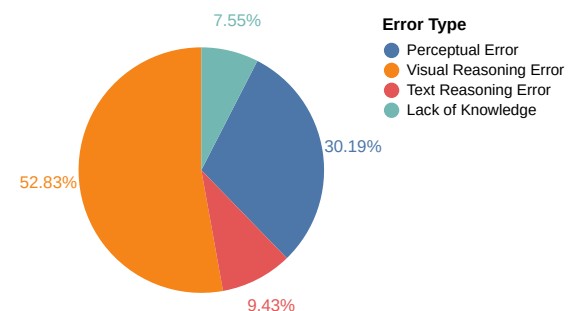

**Figure 5: Distribution of error types made by o1** on the math and coding portions of EMMA-mini. The majority of errors arise in visual reasoning.

On the other hand, we find that self-reward modeling often underperforms. Even using Gemini 2.0 Flash Thinking for self-reward modeling yields performance consistently below that of majority voting. We conjecture that self-reward modeling may be less effective because the model's evaluation criteria may be disrupted by its own generation patterns, making it less sensitive to differences in the reasoning of the generated responses than an independent reward model.

## 5. Error Analysis

**Error Distribution** We present an analysis of the errors made by o1 on the math and coding portions of EMMA-mini. In total, o1 incorrectly answers 59 math questions and 47 coding questions. Figure 5 categorizes these errors into four types. **Perceptual errors**, such as misinterpreting visual information, account for 30.19% of all errors. **Lack of knowledge errors**, including mistakes related to API usage, contribute 7.55%. **Visual reasoning errors**, such as failures to simulate 3D processes, constitute the largest category at 52.83%. Finally, **textual reasoning errors**, including calculation mistakes or logical missteps, represent 9.43%. The predominance of visual reasoning errors underscores the limitations of current models in addressing complex visual reasoning tasks. Figure 6 illustrates a representative case: while o1 correctly identifies that the problem calls for the application of the Right-Hand Rule, it fails to simulate where the thumb would point to when the right-hand fingers are curled in the direction of the current. We provide more error cases in Appendix D.

**The Effects of Textual CoT** Table 2 reveals a notable discrepancy: while CoT prompting improves reasoning in closed-source MLLMs, it tends to hurt performance in open-source models. Although the cause remains unclear without access to training data or methods, we analyze error rate per skill category to propose a conjecture. Figure 7 compares accuracy differences between CoT and Direct prompting for six models on two math tasks: 2D Transformation and Multi-Hop Object Counting. On 2D Transformation, all but one model shows reduced performance with CoT; on Multi-Hop Object Counting, while CoT generally helps closed-source

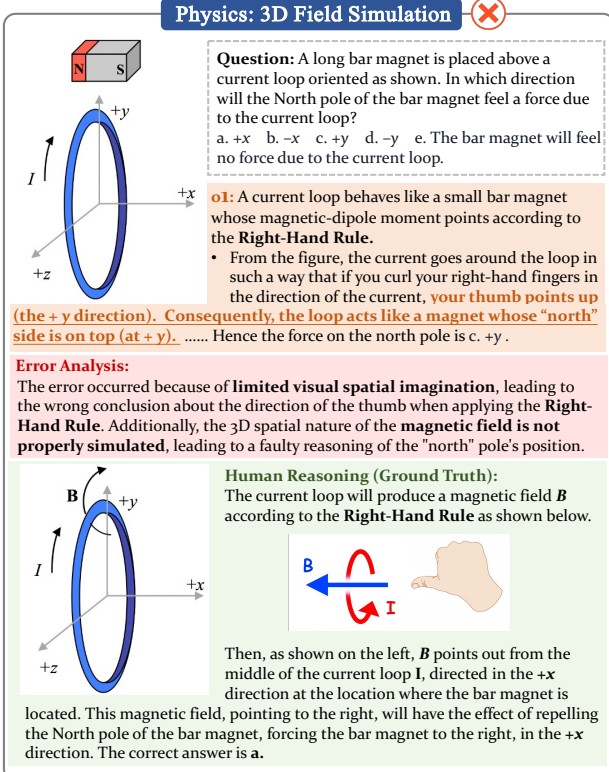

**Figure 6: A representative example of visual reasoning error** from o1. o1 misapplies the Right-Hand Rule due to limited visual spatial simulation skills.

models, it makes all but one open-source model perform worse. In fact, we notice that CoT prompting introduces more hallucinations for open-source models.

2D Transformation questions primarily test for visual simulation and spatial imagination, which are difficult to verbalize. The question in Figure 8, for example, calls for spatial imagination beyond the power of language. In contrast, Multi-Hop Object Counting can leverage language to describe the relative positions of objects. Hence, we conjecture that visual-centric tasks, such as 2D Transformation, are poorly suited for textual CoT, whereas tasks that benefit from text-based reasoning, such as Multi-Hop Object Counting, allow models to achieve greater performance gains with textual CoT, as evidenced by the closed-source models.

## 6. Conclusion

We contribute EMMA, an Enhanced MultiModal reAsoning benchmark. EMMA features multimodal questions that cannot be solved by independently reasoning within each modality. Evaluation of ten MLLMs reveals a substantial performance gap compared to human experts on EMMA, with techniques such as Chain-of-Thought prompting and test-time compute scaling offering only marginal gains. EMMA highlights the need for new architectures and training paradigms that can better integrate and rea-

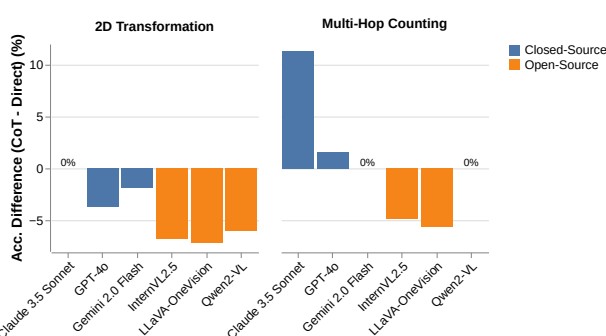

**Figure 7: Performance differences w/ and w/o CoT** for two math tasks: 2D Transformation and Multi-Hop Object Counting.

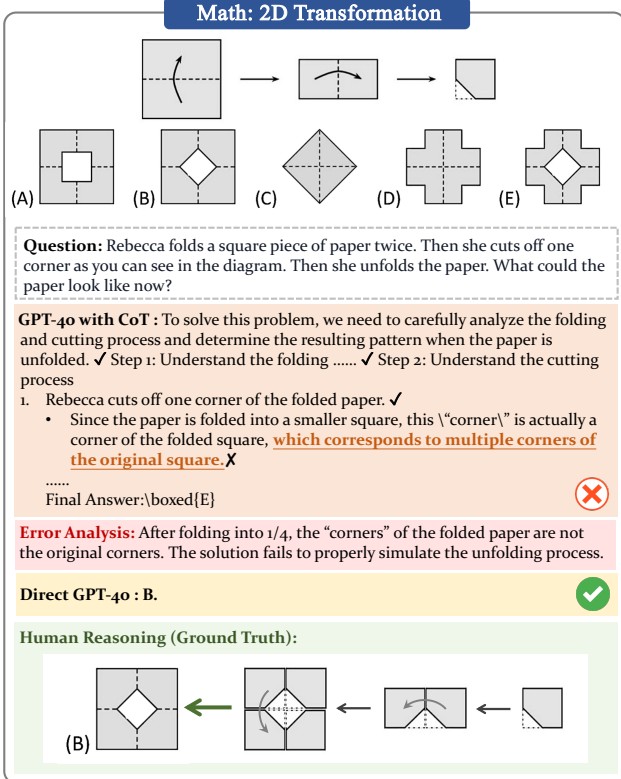

**Figure 8: An example where textual CoT is unhelpful for solving the problem.** While GPT-4o correctly solves the problem without CoT, it answers incorrectly with CoT. The thought process demonstrates a superficial association with spatial manipulations and relations rather than genuine visual reasoning.

son over diverse modalities. Like any benchmark, EMMA has its limitations, which can be improved in future works. Future iterations could also enrich the currently underrepresented physics section or expand the chemistry section to incorporate a broader range of chemistry topics. Nonetheless, EMMA sets a new standard for assessing MLLMs on multimodal reasoning.

## Impact Statement

This work addresses the challenge of evaluating multimodal reasoning in AI systems by introducing EMMA, a benchmark designed to assess models' ability to integrate textual and visual information. By curating a diverse set of difficult problems, EMMA provides a rigorous evaluation framework for understanding model performance in multimodal tasks.

The societal implications of this work are significant. Improved multimodal reasoning could enhance AI applications in science, engineering, and education, enabling more effective problem-solving and decision-making. However, our findings highlight persistent gaps in current models' ability to perform fine-grained visual reasoning, limiting their reliability in high-stakes applications.

While our curation process effectively filters out shallow multimodal reasoning, manual selection may introduce unintended biases in question formulation and domain representation. Additionally, as AI systems improve at multimodal reasoning, there is potential for misuse in generating deceptive or misleading visual content, raising ethical concerns. Addressing these challenges requires continued research into more diverse and automated benchmark construction methods, as well as responsible AI deployment.

By exposing critical weaknesses in state-of-the-art models, EMMA sets a new standard for evaluating multimodal reasoning, driving future research toward more advanced AI systems.

## Acknowledgements

We sincerely thank the human experts from Tsinghua University, Peking University, Fudan University, Nanjing University, University of Electronic Science and Technology of China, Sun Yat-sen University, Northwestern Polytechnical University, and University of Science and Technology of China for their valuable support and professional feedback throughout our work.

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

# A. Overview of the Appendix

This Appendix is organized as follows:

- Section B contains details about the composition of EMMA, the data curation process, and comparison with existing benchmarks;

- Section C contains experimental details, including the prompts used, models tested, hyperparameter settings, and the breakdown results on different categories;

- Section D contains additional case studies for each subject.

# B. EMMA Details

## B.1. Composition of EMMA

EMMA comprises 2,788 questions across four subjects: math, physics, chemistry, and coding. We now provide a detailed breakdown of EMMA by subject:

**Math** The math portion of EMMA consists of 892 questions, of which 562 are multiple-choice questions and 330 are free-form questions. These questions can be categorized into five areas: **2D Transformation** (266 questions); **3D Spatial Simulation** (275 questions); **Path Tracing/Change of View simulation** (127 questions); **Multi-Hop Visual Object Counting** (124 questions); and **Pattern Inference** (100 questions).

Tasks in the **2D Transformation** category often involve operations such as rotating, translating, or flipping shapes. Examples are provided in Figure 9. Humans typically solve these problems by leveraging their ability to "see" and mentally manipulate objects, simulating spatial transformations to arrive at a solution. During data filtering, we observe that models also rely heavily on visual information to solve these problems, as they often fail when provided only with textual descriptions of the accompanying images. Similarly, problems in the **3D Spatial Simulation** category require a similar visual reasoning approach, but with the key difference that the simulation must be performed in three-dimensional space. The **Path Tracing/Change of View Simulation** category involves solving problems akin to maze navigation, where the task requires tracing a path from a starting point to an endpoint while considering changes in perspective. We present two typical examples in Figure 10. Problems in the **Multi-Hop Visual Object Counting** category are sampled from Math-Vista (Lu et al., 2024b), with some examples shown in Figure 11. Unlike straightforward object counting, which might ask, "How many objects are present?", these questions require models to identify objects based on their attributes and perform subtraction operations grounded in visual properties. The **Pattern Inference** category involves

identifying how shapes or colors evolve across a series of diagrams and predicting the next pattern in the sequence. Solving such problems draws on the ability to recognize visual regularities, which are challenging to describe accurately using text alone, necessitating strong visual reasoning. Typical examples are provided in Figure 12.

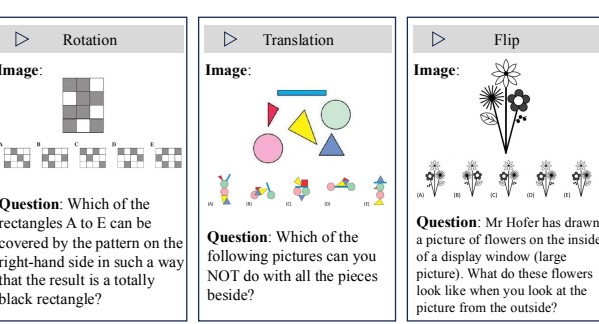

**Figure 9:** Three main types of questions belonging to the 2D Transformation category in math: Rotation, Translation, and Flipping.

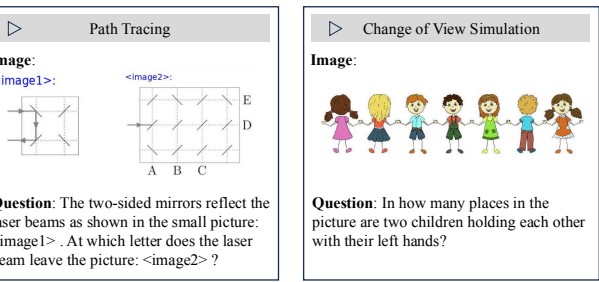

**Figure 10:** Two typical examples of the Path Tracing / Change of View Simulation category in math.

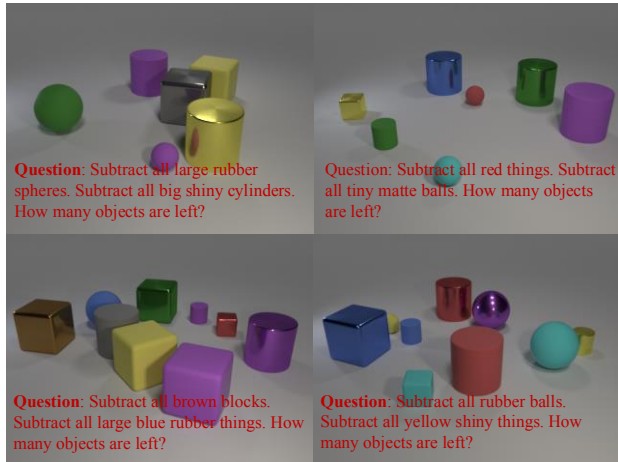

**Figure 11:** Four typical problems from the Multi-Hop Visual Object Counting category in math.

**Physics** The physics portion of EMMA includes 156 questions divided into five categories: **3D Field Simulation** (37 questions), **Graph Reasoning** (26 questions), **Multi-Hop Visual Reasoning** (33 questions), **Path Tracing** (13 questions), and **Visual Decomposition Simulation** (47 ques-

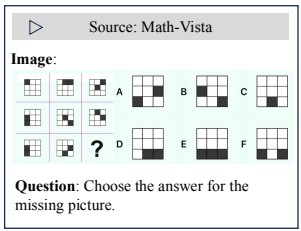
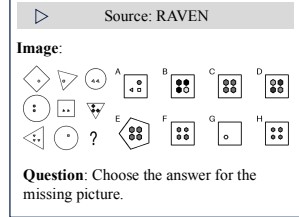

**Figure 12:** Questions in the Pattern Inference category for math are sourced from two datasets: Math-Vista (Lu et al., 2024b) and RAVEN (Zhang et al., 2019). Two examples from these sources are presented for illustration.

tions). All physics questions in EMMA are multiple-choice questions.

The **Visual Decomposition Simulation** category (47 questions) is often related to topics in physics such as Dynamics, Circular Motion, & Gravitation. It typically involves the analysis of forces acting on static or dynamic objects, which requires visual decomposition of forces and visual simulation of current and future states. The **3D Field Simulation** category (37 questions) addresses topics like Electric Force, Field, and Potential. These tasks emphasize visual simulations of properties and phenomena in three-dimensional electric and magnetic fields. The **Graph Reasoning** category (26 questions) involves interpreting and reasoning about physics-related graphs, such as velocity-time (v-t) graphs, displacement-time (s-t) graphs, and trajectory graphs. The **Multi-Hop Visual Reasoning** category (33 questions) is a mixed category, which includes some circuit analysis diagrams, as well as various types of problems that require multi-hop thinking. In the **Path Tracing** category (13 questions), most problems involve light refraction. These tasks require analyzing the paths of particles by tracing their trajectories based on the perspectives depicted in the images.

**Chemistry** The chemistry portion of EMMA includes 1,176 questions. Only 20 of them come from existing benchmarks, **the vast majority of 1,156 are created by us.** These questions can be categorized into five areas: **Knowledge-based Counting** (456 questions); **Graph Reasoning** (9 questions); **Structure Recognition** (474 questions); **Reaction Simulation** (132 questions); and **Reaction Simulation Pro** (105 questions; Based on the test Reaction Simulation skill, the requirements for visual reasoning are more professional, where all options in multiple-choice questions are images). Among the chemistry questions, only the answer corresponding to the category Knowledge-based Counting is free-form, and all the remaining questions are multiple-choice.

EMMA prioritizes chemistry problems that require rich multimodal reasoning over simple fact recall or direct applications. To fill the gaps in current datasets, we manually developed a new test suite concentrating on organic chem-

istry because we found that this type of problem makes up the majority of the remaining data set filtered. In these newly developed sections, we focus on "arrow-pushing" diagrams, a common representation used to illustrate electron flow in mechanistic steps. The dataset includes structural molecular identifiers of molecular images in chemical reaction mechanisms. The questions are categorized into three types in increasing reasoning difficulty: **Knowledge-based Counting**, **Structure Recognition**, and **Reaction Simulation**.

The **Knowledge-based Counting** category (456 questions) involves counting the number of chemical bonds of a chemical structure. The task requires domain-specific knowledge and multiple inference steps to accurately count different bonds. **Structure Recognition** (474 questions) presents a more difficult task. It requires correctly identifying the number and type of atoms and chemical bonds, recognizing the molecular structure, and deriving the corresponding chemical expression or SMILES expression. Notably, each molecule in the diagram corresponds to a unique SMILES expression, encapsulating both compositional and structural information.

**Reaction Simulation** (132 questions) is the task requiring the most advanced visual reasoning. It involves inferring the post-reaction SMILES/Chemical expression based on the molecular composition and structural information available before the reaction, guided by the direction of electron flow indicated by arrows. Due to its complexity and the poor performance of models on the simpler open-ended tasks, this task is presented in the form of multiple-choice questions. **Reaction Simulation Pro** (105 questions) assess reaction simulation skills and visual reasoning with a higher level of expertise. Each question presents options in the form of images, requiring participants to not only simulate reactions but also engage in complex multi-step reasoning by comparing different choices. **Graph Reasoning** (9 questions) involves graph-based reasoning problems related to chemistry knowledge, such as reaction rate changes.

**Coding** Implementing user interfaces (UIs) is a fundamental task in software engineering. In this work, we focus on a critical aspect of UI development: data visualization. Creating data visualizations not only requires working knowledge of the charting toolkits, but also demands reasoning over how various visual elements coordinate to achieve the desired results. To stress-test MLLMs' visualization skills, we design four tasks: **Visualization Choose Code** (188 questions; given an image of a visualization, choosing which visualization program generates it), **Code Choose Visualization** (188 questions; given a visualization program, choosing which image it generates), **Modify without the Original Image** (94 questions; given a target visualization image and a visualization program that does not yet

generate the target image, choosing what change should be applied to the program to create the target image), and **Modify with the Original Image** (94 questions; given a target visualization image, a visualization program that does not yet generate the target image, and the image that the current program generates, choosing what change should be applied to the program to create the target image). The four examples in Figure 15 illustrate each task. All coding questions in EMMA are multiple-choice questions **created by us from scratch**. Notably, these tasks simulate various real-world applications of MLLMs, requiring skills essential for replicating a target visualization or redesigning an existing one. To ensure familiarity, all visualization code in our benchmark is generated in Python using matplotlib or seaborn.

Further, similar to the other subjects, we provide fine-grained categorizations for each coding question based on the skills it measures. Since visualizations involve multiple design choices and each problem includes at least four visualizations (or visualization programs), each question may be assigned to multiple categories. On average, each question is associated with 2.11 categories. Through manual coding, we identify a total of ten categories: **3D** (108 questions; reasoning about visualizations in 3D), **Color & Texture** (156 questions; reasoning about the colors and textures of marks), **Data Reasoning** (108 questions; reasoning about the data in visualizations); **Advanced Chart Type** (276 questions; involving advanced chart types, such as fishbone diagrams), **Alignment, Orientation, & Position** (180 questions; reasoning about how visual elements should be arranged), **Gridline** (60 questions; reasoning about the use of gridlines), **Polar Coordinates** (48 questions; reasoning about charts in polar coordinates), **Axis & Scale** (108 questions; reasoning about the use of axes and scales), **Legend** (96 questions; reasoning about the appearance, content, and position of legends), **Marker & Line** (48 questions; reasoning about the style of markers and lines). Section D contains sample questions for some of the categories.

## B.2. Additional Data Curation Details

We now provide additional details on data curation for chemistry and coding.

**Chemistry** We generate ten responses with state-of-the-art LLMs for image-captioned versions of chemistry questions in existing multimodal reasoning datasets (Yue et al., 2024a; Das et al., 2024; Lu et al., 2022). Questions that are answered correctly in at least five out of ten rounds are filtered out. Analysis of the remaining questions shows that most involve molecular formulas, indicating that molecular-related tasks often require additional visual information for effective reasoning.

Based on this observation, we construct the chemistry section in EMMA from scratch, which features three tasks of increasing difficulty. The original data is sourced from SMi-CRM (Leung et al., 2024). Notably, the correct answers for the Reaction Simulation task—the most challenging and the one that best reflects vision's role in the reasoning process—are constructed and verified by a PhD candidate in chemical molecules. The questions and answers for the Reaction Simulation task are sourced from Li et al.'s collection of chemical reactions (Li).

**Coding** We manually curate all questions for coding. Our curation process consists of three stages. In the first stage, we identify "seed visualizations" that employ advanced visualization techniques or present a rich space for design variations. We source these seed visualizations through three channels: CharXiv (Wang et al., 2024c) (a benchmark consisting of diverse charts extracted from arXiv papers), the official matplotlib example gallery [2], and our prior experience. For each source, we attempt to reproduce visualizations demonstrating advanced techniques (e.g., 3D bar charts) in Python using GPT-4o or Claude 3.5 Sonnet, retaining only those that MLLMs cannot reasonably replicate after multiple iterations of prompting. The first stage ultimately results in 47 seed visualizations.

In the second stage, we generate four variations for each seed visualization. When we prompt MLLMs to reconstruct visualizations during the first stage, MLLM-generated visualizations are often ill-formed, nonsensical, or otherwise fail to achieve the desired effects. However, since these visualizations are generated by MLLMs, they may be indistinguishable from the correct visualizations to the models. Yet, it is crucial for MLLMs to recognize such flaws, as early identification of errors is essential for efficient human-AI collaboration. As such, we include such "buggy" code snippets as variations of seed visualizations. In addition, we further enrich the set by introducing design variations (e.g., changes in spine configuration, line style, or axis scaling) either manually or through prompting MLLMs, with post-hoc manual verification. After the second stage, we are left with 188 visualizations, organized into 47 sets, each containing four visualizations.

In the third stage, we construct questions using these 188 visualizations. For Vis Choose Code, we iterate through each visualization within a set and construct questions asking models to select the code snippet used to generate the chart. For Code Choose Vis, we iteratively choose a code snippet from each set and ask models to identify the corresponding generated image. For Modification, we first introduce another design variation in each set, and then select two

---

[2] https://matplotlib.org/stable/gallery/index.html. This approach is inspired by Wu et al. (Wu et al., 2024).

pairs of visualizations from the set, where each set contains a relatively well-formed chart and another random chart. We construct questions by comparing the code of the randomly selected snippet with others, asking what changes are needed to produce the target visualization. While the target visualization image is requisite, we vary whether the original visualization image is provided. In sum, this procedure generates four Code Choose Vis questions, four Vis Choose Code questions, and four Modification questions (two with the original image and two without) per set, resulting in a total of 564 questions evenly divided among the tasks.

### B.3. Comparison with Other Benchmarks

EMMA stands out from existing multimodal benchmarks by emphasizing questions that truly demand multimodal reasoning capabilities. Through meticulous manual labeling or verification, we provide fine-grained labels for each question, categorizing them based on the specific skills they assess. This approach enables a more detailed analysis of the limitations of MLLMs.

**Math** Various benchmark datasets (Lu et al., 2024b; Wang et al., 2024a; Yue et al., 2024a;b) have been proposed to evaluate the mathematical reasoning capabilities of MLLMs. However, existing math benchmarks often emphasize shallow perceptual cues or rely heavily on text-dominant reasoning. In contrast, our dataset mainly focuses on assessing the performance of MLLMs on tasks that require integrated reasoning, particularly those that are highly dependent on visual information. Specifically, we employ an enhanced data filtering pipeline to separate questions that could be answered correctly using only the caption of images. Representative examples of such problems are illustrated in Figure 13. In some cases, the images provide no additional information required to solve the question, and the answer can be derived entirely from the text of the question alone, as shown in the middle example in Figure 13. In other instances, questions can be solved using image captions, leveraging the text reasoning capabilities of MLLMs. These images either consist solely of textual information, as illustrated in the rightmost example in Figure 13, or can be fully described textually without necessitating further reasoning involving image transformations. For instance, in the leftmost example in Figure 13, as long as the key textual information in the image, such as $y = 0.5^x$, is identified, the question can be solved without employing a graphical approach.

In addition to the above, we provide a fine-grained taxonomy for math problems. By first categorizing the questions using GPT-4o and conducting expert-level manual verification of the classifications, we identify categories that are highly likely to require graphic transformation and spatial simulation. These categories are not only applicable to all

math problems in our dataset but are also adaptable to other dataset, such as MMMU (Yue et al., 2024a). In figure 14, we present two examples from MMMU that fall under our defined categories. We hope these categories will inspire further exploration of the visual reasoning capability of MLLMs.

**Science (Physics & Chemistry)** The latest multimodal reasoning benchmarks in science, such as (Yue et al., 2024a; Das et al., 2024; Lu et al., 2022), do not provide many multimodal physics and chemistry problems. In addition, they often focus on superficial visual cues or heavily rely on text-based reasoning. As pointed out by (Fu et al., 2024), text representation can address 90% of physics and chemistry questions in ScienceQA (Lu et al., 2022). As a result, our filtering pipeline leaves only 100 problems in total for these subjects from relevant benchmarks, which we expand to 1,332 with our newly constructed problems.

Our benchmark places greater emphasis on the role of vision in multimodal reasoning. Beyond filtering out problems solvable through text alone, we manually review and annotate the remaining questions to ensure a strong reliance on visual information.

In particular, we focus on two specific types of problems. First, some physics problems require visual imagination and simulation of physical processes. Auxiliary images can significantly enhance both the accuracy and efficiency of problem-solving. Second, in chemistry, tasks such as molecule counting, structure recognition, and reaction simulation demand effective utilization of visual information. These two types of problems and their related data have been largely overlooked in current science datasets. To address this gap, we emphasize these under-explored aspects in the science portion of EMMA by manually constructing test questions and carefully sourcing them from existing datasets.

**Coding** Most current visualization-related benchmarks assess visualization understanding (Masry et al., 2022; Xia et al., 2024; Liu et al., 2022; Methani et al., 2020). In this work, we focus on evaluating how well MLLMs can reason in multimodality when *generating* visualizations. To this end, past work (Han et al., 2023; Wu et al., 2024; Shi et al., 2024; Zhang et al., 2024b; Hu et al., 2024) has proposed the task of visualization reproduction—generating code to reproduce a target visualization. To evaluate the quality of generations, researchers have developed heuristic measures and employed MLLMs as judges.

EMMA enhances past work by introducing new task types. Our four tasks enable targeted assessments of *Vis2Code*, *Code2Vis*, and *Visualization Modification*. In particular, while many users rely on MLLMs for visualization debugging, this task has not been addressed by existing bench-

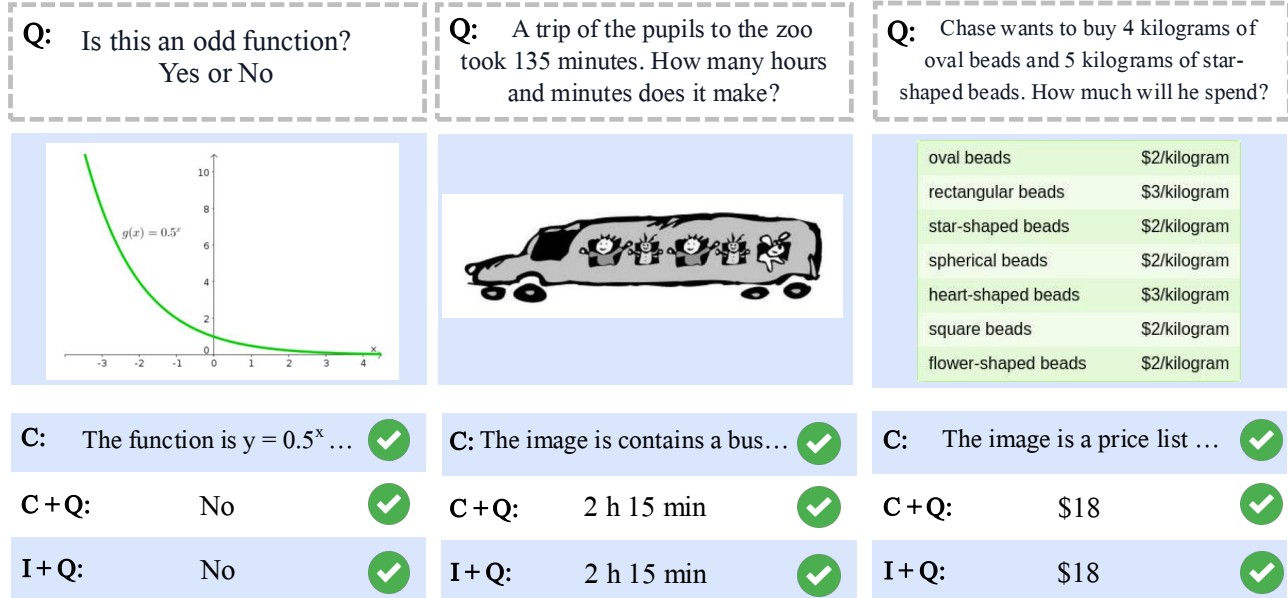

**Figure 13:** Three typical examples from other math datasets that do not really require the images or can be correctly answered using only the captions of images. The leftmost and rightmost examples are from Math-Vista (Lu et al., 2024b) and the middle example is from Math-Vision (Wang et al., 2024a). **Q** represents the question, **C** represents the caption of the image, and **I** represents the image.

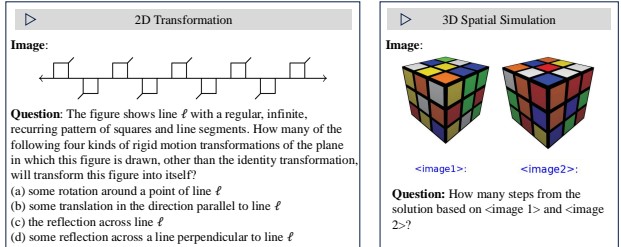

**Figure 14:** Two examples from MMMU (Yue et al., 2024a), which belong to our defined categories of 2D Transformation and 3D Spatial Simulation respectively. Our classification is equally applicable to other datasets.

marks. We further provide fine-grained, expert-generated categorizations for each question based on the skills it measures (see Section B.1). Finally, all of our coding questions are posed as multiple-choice questions, which removes the need for using MLLMs as judges, which can be unreliable.

## C. Experimental Details

### C.1. Prompts for Data Curation

As previously discussed, we conduct much filtering to cull out questions from existing datasets that genuinely require visual reasoning, which involves using GPT-4o to generate captions for images and passing the captions along with textual questions to MLLMs to generate responses. The prompts used to generate captions and responses are shown in Table 4. Notably, when generating image captions, we also provide the corresponding questions to models to make sure the captions are as accurate as possible.

For math, after filtering the data, we conduct a detailed observation and analysis of the remaining problems and develop a taxonomy consisting of five categories. We then utilize GPT-4o to assist with the categorization and the prompts used during this process are shown in Table 4.

### C.2. Prompts for Response Generation

We evaluate several state-of-the-art MLLMs on EMMA, considering two different prompting strategies: *Direct* prompting and Chain-of-Thought (*CoT*) prompting. Furthermore, our benchmark features two question types: *Open-ended* and *Multiple Choice*. The corresponding prompts vary according to the type of question and the prompting strategy, as shown in Table 5

### C.3. Models and Settings

During math data curation, we filter out questions that models can answer when provided only with the image caption and the question. In this process, each model generates ten candidate responses to ensure reliable and effective filtering. To expedite response generation, we use the vLLM (Kwon et al., 2023) library, an open-source tool for fast LLM inference and serving. For all other cases, we load models directly using the Transformers (Wolf et al., 2020) library. All model sources are official and listed in Table 6. When evaluating different models, we use default hyperparameter values unless otherwise specified, with detailed parameter settings provided in Table 6.

**Table 4:** The prompts used to caption images and generate responses during data curation.

| Setting | Prompt |
|---------|--------|
| Generate Captions | There is a question about the image or figure. Please describe the fine-grained content of the image or figure based on this question, including scenes, objects, relationships, and any text present. Please note that you do not need to answer this question directly, just describe the information of this picture. |
| Generate Responses | Please first solve the problem step by step, then put your final answer or a single letter (if it is a multiple choice question) in one "\boxed{}". Here is the natural description of the figure, please solve the following problem based on the description. |
| Categorize | There are some math problems combining images and text, and existing large models cannot correctly reason through these problems. We have analyzed the reasons why large models fail to classify these problems and have categorized them based on the challenges present in the problems. The categories are as follows: A: To solve the problem, a 2D transformation is required, such as translation, rotation, scaling, shearing, reflection, etc. B: To solve the problem, 3D spatial imagination is needed. C: To solve the problem, path tracing/change of view simulation is needed, such as a math problem about a maze. D: None of the above. It belongs to another category. Here is a math problem, please give the math category that you think this problem belongs to and explain why. If you choose D, please additionally include the type of math problem you believe it to be. |

**Table 5:** The prompts used for evaluation across different question types and prompting strategies.

| Type | Strategy | Prompt |
|------|----------|--------|
| Open-ended | CoT | {context}{question}
Answer the question using a single word or phrase and put the answer in one "\boxed{}". Please solve the problem step by step. |
| | Direct | {context} {question}
Answer the question using a single word or phrase and put the answer in one "\boxed{}". Please ensure that your output only contains the final answer without any additional content (such as intermediate reasoning steps). |
| Multiple Choice | CoT | {context} {question} {options}
Answer with the option's letter from the given choices and put the letter in one "\boxed{}". Please solve the problem step by step. |
| | Direct | {context} {question} {options}
Answer with the option's letter from the given choices and put the letter in one "\boxed{}". Please ensure that your output only contains the final answer without any additional content (such as intermediate reasoning steps). |

**Table 6:** The sources of models used in the experiments and the hyperparameters configuration. Pass@1 refers to scenarios where evaluation is performed only once, while pass@n refers to cases that require generating multiple candidate responses.

| Model | Parameter Setting | Source | URL |
|---|---|---|---|
| GPT-4o pass@1 | temperature = 0.0 | chatgpt-4o-latest | https://platform.openai.com |
| GPT-4o pass@n | temperature = 0.7 | chatgpt-4o-latest | https://platform.openai.com |
| Claude 3.5 Sonnet | temperature = 0.0 | claude-3-5-sonnet | https://www.anthropic.com/ |
| Gemini 2.0 Flash pass@1 | temperature = 0.0 | gemini-2.0-flash-exp | https://ai.google.dev/ |
| Gemini 2.0 Flash pass@n | temperature = 0.7 | gemini-2.0-flash-exp | https://ai.google.dev/ |
| Gemini 2.0 Flash Thinking-1219 | temperature = 0.0 | gemini-2.0-flash-thinking-exp-1219 | https://ai.google.dev/ |
| Gemini 2.0 Flash Thinking-0121 | temperature = 0.0 | gemini-2.0-flash-thinking-exp-0121 | https://ai.google.dev/ |
| OpenAI o1 | - | interface | https://chatgpt.com/ |
| Qwen2-VL-72B-Instruct | temperature = 0.7 | local checkpoint | https://huggingface.co/Qwen/Qwen2-VL-72B-Instruct |
| QVQ-72B-Preview | temperature = 0.7 | local checkpoint | https://huggingface.co/Qwen/QVQ-72B-Preview |
| LLaVA-Onevision-72B | do_sample=True, temperature = 0.7 | local checkpoint | https://huggingface.co/llava-hf/llava-onevision-qwen2-72b-ov-hf |
| InternVL2-Llama3-76B | do_sample=True, temperature = 0.7 | local checkpoint | https://huggingface.co/OpenGVLab/InternVL2-Llama3-76B |
| InternVL2.5-78B | do_sample=True, temperature = 0.7 | local checkpoint | https://huggingface.co/OpenGVLab/InternVL2_5-78B |

**Table 7:** Test-time scaling results on the math portion of EMMA-mini using Qwen2.5-Math-RM-72B, a specialized reward model for math, and some generalist models as reward models. Since Qwen2.5-Math-RM-72B does not take images as input, we provide GPT-4o-generated captions of images in the problems to it. Overall, generalist reward models provide better reward signals than Qwen2.5-Math-RM-72B.

| Model | Method | N=1 | N=2 | N=4 | N=8 | N=16 |
|---|---|---|---|---|---|---|
| GPT-4o | BoN w. Self-RM | | 29.00 | 27.00 | 25.00 | — |
| | BoN w. Gemini 2.0 Flash Thinking as RM | 27.00 | 30.00 | 28.00 | **31.00** | — |
| | BoN w. Qwen2.5-Math-RM-72B as RM | | 26.00 | 24.00 | 25.00 | 29.00 |
| Gemini 2.0 Flash | BoN w. Self-RM | | **33.00** | 24.00 | 27.00 | — |
| | BoN w. Gemini 2.0 Flash Thinking as RM | 24.00 | **33.00** | 24.00 | 25.00 | — |
| | BoN w. Qwen2.5-Math-RM-72B as RM | | 27.00 | 28.00 | 23.00 | 23.00 |
| Gemini 2.0 Flash Thinking | — | **35.00** | — | — | — | — |
| o1 | — | **41.00** | — | — | — | — |

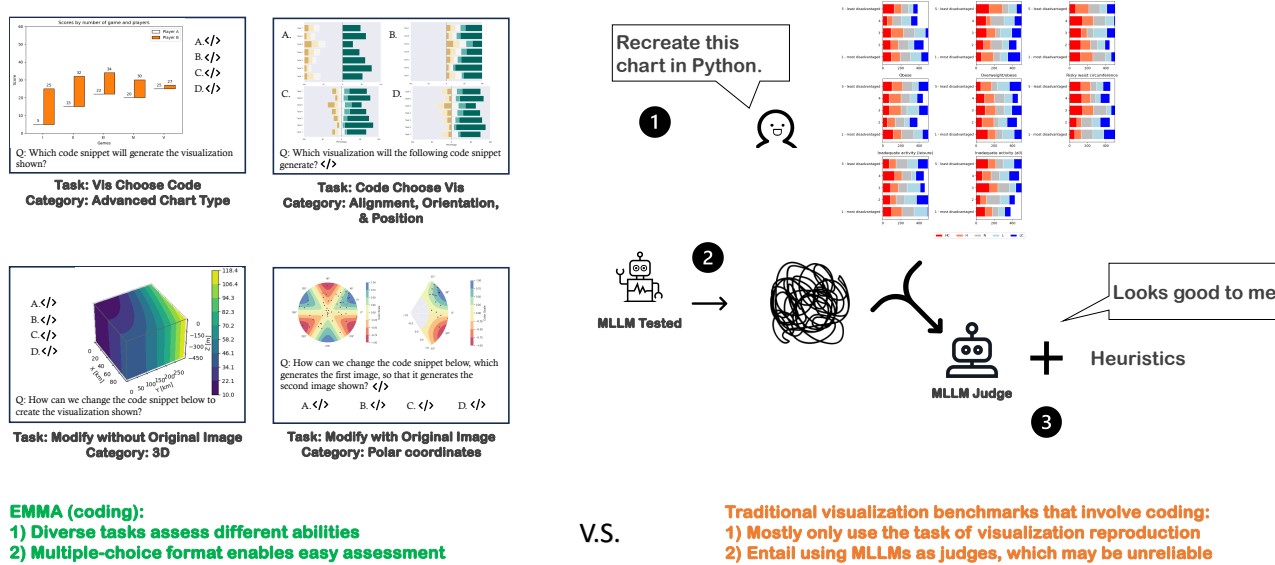

**Figure 15:** Traditional visualization benchmarks for coding often require MLLMs to recreate a target visualization, with evaluations conducted using a combination of MLLMs as judges and heuristic methods. In contrast, EMMA introduces four visualization-related coding tasks designed to assess multimodal coding abilities across multiple dimensions. By employing a multiple-choice format, EMMA eliminates the reliance on potentially unreliable MLLM-based judgment. Additionally, our fine-grained categories facilitate a detailed analysis of the limitations of multimodal coding skills.

## C.4. Best-of-N With a Specialized Math Reward Model

While we want to evaluate specialized reward models in addition to generalist reward models, they are currently only available for math. Qwen2.5-Math-RM-72B (Yang et al., 2024b) is one such specialized LLM for evaluating the quality of responses to math problems. For the math portion of EMMA-mini, we first generate 16 responses using each of GPT-4o and Gemini 2.0 Flash. We then score each response using Qwen2.5-Math-RM-72B and select the highest-scoring answer. As in previous experiments, we test with $N$ values of 2, 4, 8, and 16. Since Qwen2.5-Math-RM-72B is not multimodal, we supply GPT-4o-generated captions for all images in the questions to enable evaluation.

Table 7 compares scaling results on the math portion of EMMA-mini using Qwen2.5-Math-RM-72B, a specialized reward model for math, against other generalist reward models. Overall, generalist reward models generate better rewards than Qwen2.5-Math-RM-72B. We note, however, that Qwen2.5-Math-RM-72B is not a text-only model, which likely affects its performance on a multimodal benchmark like EMMA.

## C.5. Breakdown of Experiment Results by Category

In this section, we present a detailed breakdown of the results for each category across different subjects. Specifically, the results for math are shown in Table 8, for physics in Table 9, for chemistry in Table 10, and for coding in Table 11.

## D. Case Study

We now present additional case studies, showcasing both correct and incorrect responses by MLLMs, organized by subject. We provide the original questions, MLLM responses, ground truth solutions, and our error analyses. Some questions also feature o1 responses to the captioned version of the original questions. Overall, we find that a recurring pattern in error cases is that MLLMs fail to fully engage their multimodal reasoning skills. When questions require multiple visual passes or extended visual simulation or manipulation, MLLMs often skip over these critical steps in their thought processes or produce completely incorrect solutions.

**Table 8:** Performance of state-of-the-art MLLMs on Math. Column abbreviations: 2D = 2D Transformation, 3D = 3D Spatial Simulation, Path = Path Tracing, Pat = Pattern Inference, MH = Multi-Hop Visual Object Counting.

| | EMMA-mini | EMMA | | | | | |
|---|---|---|---|---|---|---|---|
| | **Overall** (100) | **2D** (266) | **3D** (275) | **Path** (127) | **MH** (124) | **Pat** (100) | **Overall** (892) |
| Random choice | 13.00 | 15.04 | 12.73 | 10.24 | 22.58 | 9.00 | 14.01 |
| Human Expert | 75.00 | - | - | - | - | - | - |
| Direct Claude 3.5 Sonnet | 23.00 | 26.69 | 18.18 | 21.26 | 49.19 | 17.00 | 25.34 |
| Direct Gemini 2.0 Flash | 20.00 | 25.19 | 20.73 | 19.69 | 37.90 | 17.00 | 23.88 |
| Direct GPT-4o | 30.00 | 27.44 | 19.64 | 17.32 | 58.87 | 21.00 | 27.24 |
| Direct Qwen2-VL-72B-Instruct | 38.00 | 24.81 | 20.00 | 18.90 | 78.23 | 53.00 | 33.07 |
| Direct LLaVA-Onevision-72B | 25.00 | 24.81 | 22.18 | 20.47 | 69.35 | 8.00 | 27.69 |
| Direct InternVL2-Llama3-76B | 31.00 | 22.18 | 14.55 | 22.83 | 65.32 | 15.00 | 25.11 |
| Direct InternVL2.5-78B | 30.00 | 28.95 | 21.82 | 18.90 | 80.65 | 19.00 | 31.39 |
| CoT Claude 3.5 Sonnet | 30.00 | 26.69 | 22.18 | 22.83 | 60.48 | 26.00 | 29.37 |
| CoT Gemini 2.0 Flash | 24.00 | 23.31 | 26.55 | 15.75 | 37.90 | 29.00 | 25.90 |
| CoT GPT-4o | 27.00 | 23.68 | 17.82 | 14.17 | 60.48 | 23.00 | 25.56 |
| CoT Gemini 2.0 Flash Thinking-1219 | 35.00 | 30.83 | 27.64 | 20.47 | 60.48 | 23.00 | 31.61 |
| CoT Gemini 2.0 Flash Thinking-0121 | 34.00 | 32.33 | 37.82 | 26.77 | 64.52 | 27.00 | 37.11 |
| CoT OpenAI o1 | 41.00 | - | - | - | - | - | - |
| CoT Qwen2-VL-72B-Instruct | 32.00 | 18.80 | 16.00 | 14.96 | 78.23 | 37.00 | 27.69 |
| CoT LLaVA-Onevision-72B | 23.00 | 19.17 | 13.45 | 16.54 | 64.52 | 11.00 | 22.42 |
| CoT InternVL2-Llama3-76B | 27.00 | 16.17 | 14.55 | 15.75 | 64.52 | 15.00 | 22.20 |
| CoT InternVL2.5-78B | 31.00 | 22.18 | 13.09 | 16.54 | 75.81 | 18.00 | 25.56 |

**Table 9:** Performance of state-of-the-art MLLMs on Physics. Column abbreviations: Path = Path Tracing, 3D = 3D Field Simulation, MH = Multi-Hop Visual Reasoning, VD = Visual Decomposition Simulation, GR = Graph Reasoning.

| | EMMA-mini | EMMA | | | | | |
|---|---|---|---|---|---|---|---|
| | **Overall** (100) | **Path** (13) | **3D** (37) | **MH** (33) | **VD** (47) | **GR** (26) | **Overall** (156) |
| Random choice | 23.00 | 38.46 | 21.62 | 27.27 | 31.91 | 11.54 | 25.64 |
| Human Expert | 64.50 | - | - | - | - | - | - |
| Direct Claude 3.5 Sonnet | 34.00 | 30.77 | 37.84 | 36.36 | 31.91 | 30.77 | 33.97 |
| Direct Gemini 2.0 Flash | 40.00 | 38.46 | 29.73 | 42.42 | 38.30 | 46.15 | 38.46 |
| Direct GPT-4o | 38.00 | 30.77 | 40.54 | 33.33 | 36.17 | 50.00 | 38.46 |
| Direct Qwen2-VL-72B-Instruct | 40.00 | 30.77 | 48.65 | 45.45 | 42.55 | 34.62 | 42.31 |
| Direct LLaVA-Onevision-72B | 32.00 | 23.08 | 27.03 | 39.39 | 48.94 | 26.92 | 35.90 |
| Direct InternVL2-Llama3-76B | 22.00 | 15.38 | 32.43 | 21.21 | 6.45 | 23.08 | 22.44 |
| Direct InternVL2.5-78B | 40.00 | 38.46 | 43.24 | 42.42 | 38.30 | 26.92 | 38.46 |
| CoT Claude 3.5 Sonnet | 38.00 | 53.85 | 37.84 | 36.36 | 42.55 | 42.31 | 41.03 |
| CoT Gemini 2.0 Flash | 41.00 | 30.77 | 29.73 | 36.36 | 44.68 | 46.15 | 38.46 |
| CoT GPT-4o | 44.00 | 69.23 | 35.14 | 39.39 | 44.68 | 46.15 | 43.59 |
| CoT Gemini 2.0 Flash Thinking-1219 | 57.00 | 61.54 | 43.24 | 57.58 | 61.70 | 61.54 | 56.41 |
| CoT Gemini 2.0 Flash Thinking-0121 | 63.00 | 61.54 | 54.05 | 60.61 | 57.45 | 73.08 | 60.26 |
| CoT OpenAI o1 | 49.00 | - | - | - | - | - | - |
| CoT Qwen2-VL-72B-Instruct | 34.00 | 15.38 | 35.14 | 33.33 | 34.04 | 46.15 | 34.62 |
| CoT LLaVA-Onevision-72B | 26.00 | 15.38 | 10.81 | 18.18 | 6.38 | 15.38 | 12.18 |
| CoT InternVL2-Llama3-76B | 33.00 | 53.85 | 27.03 | 21.21 | 12.90 | 38.46 | 32.05 |
| CoT InternVL2.5-78B | 36.00 | 46.15 | 35.14 | 30.30 | 46.81 | 42.31 | 39.74 |

**Table 10:** Performance of state-of-the-art MLLMs on Chemistry. Column abbreviations: SR = Structure Recognition, GR = Graph Reasoning, RS = Reaction Simulation, RS-pro = Reaction Simulation-Pro, KS = Knowledge-based Counting.

| | EMMA-mini | EMMA | | | | | |
| --- | --- | --- | --- | --- | --- | --- | --- |
| | Overall (100) | SR (474) | GR (9) | RS (132) | RS-pro (105) | KS (456) | Overall (1176) |
| Random choice | 27.00 | 24.47 | 33.33 | 27.27 | 35.24 | 0.44 | 16.50 |
| Human Expert | 86.00 | - | - | - | - | - | - |
| Direct Claude 3.5 Sonnet | 44.00 | 66.88 | 22.22 | 55.30 | 55.24 | 6.80 | 40.90 |
| Direct Gemini 2.0 Flash | 36.00 | 54.01 | 11.11 | 53.79 | 58.10 | 8.33 | 36.31 |
| Direct GPT-4o | 33.00 | 47.05 | 11.11 | 51.52 | 42.86 | 8.33 | 31.89 |
| Direct Qwen2-VL-72B-Instruct | 34.00 | 45.99 | 33.33 | 48.48 | 45.71 | 9.65 | 32.06 |
| Direct LLaVA-Onevision-72B | 24.00 | 38.19 | 33.33 | 39.39 | 26.67 | 7.24 | 25.26 |
| Direct InternVL2-Llama3-76B | 21.00 | 37.34 | 22.22 | 31.45 | 24.76 | 8.55 | 24.06 |
| Direct InternVL2.5-78B | 38.00 | 55.06 | 33.33 | 47.73 | 43.81 | 8.99 | 35.20 |
| CoT Claude 3.5 Sonnet | 41.00 | 57.17 | 33.33 | 58.33 | 58.10 | 15.57 | 41.07 |
| CoT Gemini 2.0 Flash | 36.00 | 22.15 | 33.33 | 50.00 | 59.05 | 11.84 | 24.66 |
| CoT GPT-4o | 35.00 | 42.41 | 33.33 | 51.52 | 45.71 | 16.67 | 33.67 |
| CoT Gemini 2.0 Flash Thinking-1219 | 41.00 | 48.31 | 33.33 | 45.45 | 69.52 | 17.76 | 37.93 |
| CoT Gemini 2.0 Flash Thinking-0121 | 47.00 | 53.16 | 66.67 | 48.48 | 58.10 | 23.25 | 41.58 |
| CoT OpenAI o1 | 40.00 | - | - | - | - | - | - |
| CoT Qwen2-VL-72B-Instruct | 32.00 | 33.33 | 11.11 | 37.12 | 42.86 | 7.89 | 24.57 |
| CoT LLaVA-Onevision-72B | 23.00 | 33.76 | 0.00 | 37.88 | 20.95 | 7.24 | 22.53 |
| CoT InternVL2-Llama3-76B | 21.00 | 29.11 | 22.22 | 30.65 | 22.86 | 6.58 | 19.73 |
| CoT InternVL2.5-78B | 24.00 | 37.13 | 33.33 | 37.12 | 33.33 | 13.16 | 27.47 |

**Table 11:** Performance of state-of-the-art MLLMs on Coding. Column abbreviations: CCV = Code Choose Vis, VCC = Vis Choose Code, MwoI = Modify without Original Image, MwI = Modify with Original Image.

| | EMMA-mini | EMMA | | | | |
| --- | --- | --- | --- | --- | --- | --- |
| | Overall (100) | CCV (188) | VCC (188) | MwoI (94) | MwI (94) | Overall (564) |
| Random choice | 28.00 | 22.87 | 23.94 | 29.79 | 30.85 | 25.71 |
| Human Expert | 85.50 | - | - | - | - | - |
| Direct Claude 3.5 Sonnet | 35.00 | 32.98 | 41.49 | 40.43 | 42.55 | 38.65 |
| Direct Gemini 2.0 Flash | 41.00 | 39.36 | 42.02 | 43.62 | 45.74 | 42.02 |
| Direct GPT-4o | 40.00 | 43.09 | 35.11 | 40.43 | 43.62 | 40.07 |
| Direct Qwen2-VL-72B-Instruct | 37.00 | 35.11 | 30.85 | 36.17 | 39.36 | 34.57 |
| Direct LLaVA-Onevision-72B | 28.00 | 22.34 | 28.19 | 38.30 | 32.98 | 28.72 |
| Direct InternVL2-Llama3-76B | 28.00 | 27.66 | 39.52 | 30.85 | 28.72 | 27.84 |
| Direct InternVL2.5-78B | 33.00 | 30.85 | 31.38 | 35.11 | 31.91 | 31.91 |
| CoT Claude 3.5 Sonnet | 39.00 | 39.36 | 38.83 | 43.62 | 43.62 | 40.60 |
| CoT Gemini 2.0 Flash | 44.00 | 38.30 | 46.28 | 37.23 | 39.36 | 40.96 |
| CoT GPT-4o | 38.00 | 40.43 | 35.11 | 44.68 | 38.30 | 39.01 |
| CoT Gemini 2.0 Flash Thinking-1219 | 41.00 | 43.62 | 46.81 | 39.36 | 40.43 | 43.44 |
| CoT Gemini 2.0 Flash Thinking-0121 | 48.00 | 47.87 | 52.13 | 45.74 | 42.55 | 38.06 |
| CoT OpenAI o1 | 53.00 | - | - | - | - | - |
| CoT Qwen2-VL-72B-Instruct | 23.00 | 31.38 | 28.72 | 25.53 | 30.85 | 29.43 |
| CoT LLaVA-Onevision-72B | 29.00 | 24.47 | 32.98 | 35.11 | 34.04 | 30.67 |
| CoT InternVL2-Llama3-76B | 32.00 | 24.47 | 29.79 | 38.30 | 35.11 | 30.32 |
| CoT InternVL2.5-78B | 19.00 | 25.53 | 25.00 | 25.53 | 24.47 | 25.18 |

**Figure 16:** A sample correct case in math.

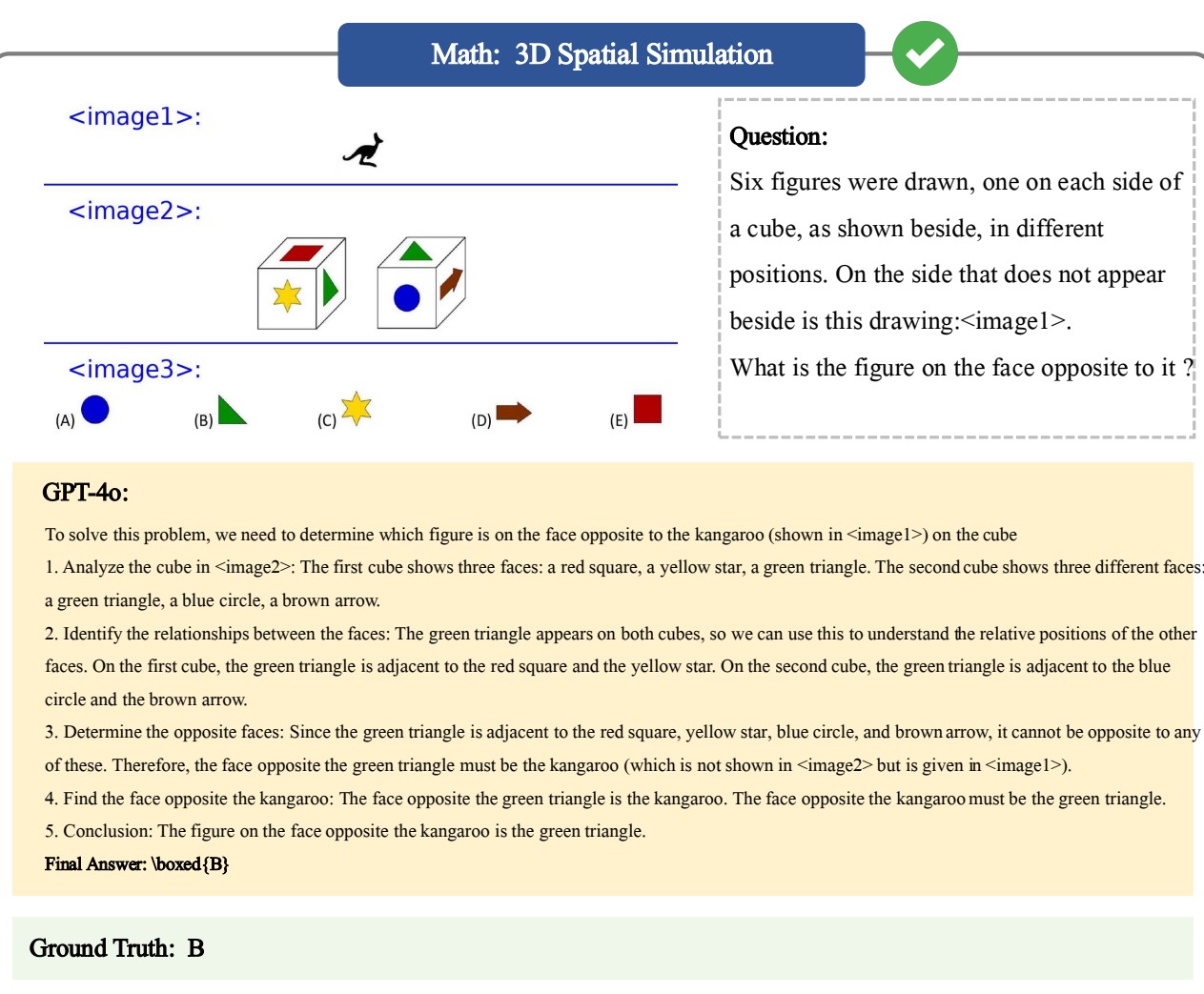

**Figure 17:** A sample correct case in math.

## Math: Path Tracing / Change of View ✅

<image1>:

<image2>:

(A) (B) (C) (D) (E)

**Question:**

Three people walked through the snow in their winter boots. In which order did they walk through the snow?

**GPT-4o:**

To solve this problem, we need to determine the order in which the three people walked through the snow based on the footprints shown in the diagram.

1. Identify the footprints: In <image1>, we see three distinct types of footprints: (A) One with a dotted pattern. (B) One with a striped pattern. (C) One with a solid sole with two ovals. In <image2>, the footprints are labeled as follows: (A) Dotted, Striped, Solid. (B) Dotted, Solid, Striped. (C) Striped, Dotted, Solid. (D) Striped, Solid, Dotted. (E) Solid, Striped, Dotted.

2. Analyze the overlapping footprints: In <image1>, we can see that some footprints overlap, which indicates that one person walked over the footprints of another. The solid sole with two ovals footprints are on top of the other two types of footprints in several places, indicating that this person walked last. The striped pattern footprints overlap the dotted pattern footprints in some areas, indicating that the person with the striped pattern walked second. The dotted pattern footprints are at the bottom, meaning this person walked first.

3. Determine the order: Based on the analysis, the order is: 1. Dotted pattern (first). 2. Striped pattern (second). 3. Solid sole with two ovals (third). 4. Match the order with the options in <image2>: The correct order is Dotted, Striped, Solid, which corresponds to option (A).

Final Answer:\boxed{A}

**Ground Truth: A**

**Figure 18:** A sample correct case in math.

## Math: Multi-hop Visual Object Counting 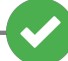

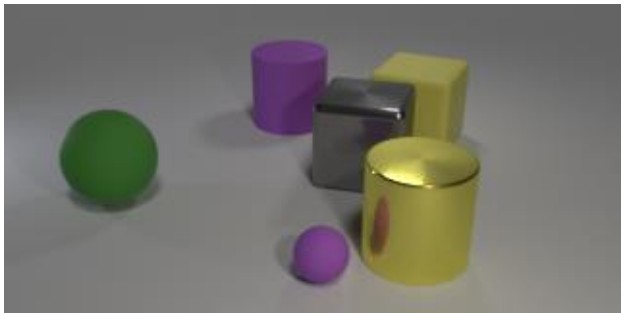

**Question:**

Subtract all large rubber spheres. Subtract all big shiny cylinders. How many objects are left?

**GPT-4o:**

Let's analyze the image step by step:

1. Identify the objects: There is a green large rubber sphere on the left. There is a purple small rubber sphere near the center. There is a purple large rubber cylinder in the back. There is a yellow large shiny cylinder on the right. There is a metallic cube in the center. There is a yellow large rubber cube in the back.

2. Subtract all large rubber spheres: The only large rubber sphere is the green sphere on the left. We subtract this object.

3. Subtract all big shiny cylinders: The only big shiny cylinder is the yellow cylinder on the right. We subtract this object.

4. Count the remaining objects: Purple small rubber sphere (still present). Purple large rubber cylinder (still present). Metallic cube (still present). Yellow large rubber cube (still present). Thus, after subtracting the specified objects, there are 4 objects left.

**Final answer: 4.**

**Ground Truth: 4**

**Figure 19:** A sample correct case in math.

**Figure 20:** A sample correct case in math.

## Math: 2D Transformation 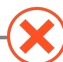

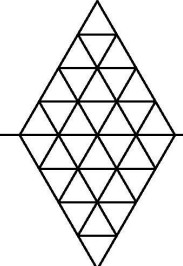

**Error Analysis:**

**Question:** Each half of this figure is composed of 3 red triangles, 5 blue triangles and 8 white triangles. When the upper half is folded down over the centerline, 2 pairs of red triangles coincide, as do 3 pairs of blue triangles. There are 2 red-white pairs. How many white pairs coincide?

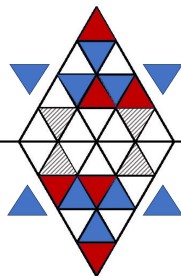

**O1:** **1.**Count the total number of triangles per half: ... **2.** When folded: ... **3.** Given pairings: ... **4.** Account for these given pairs: ... **5.** Update the counts after using these pairs: ... **6.** Remaining pairs to form: ... The remaining top and bottom sets are the same: 2B + 7W on top and 2B + 7W on bottom. **7. Forming the remaining pairs: With equal numbers of blues and whites on both sides, the most natural arrangement is to pair like colors with like colors:- Pair the 2 remaining top B with the 2 remaining bottom B → 2 more B-B pairs. - Pair the 7 remaining top W with the 7 remaining bottom W → 7 W-W pairs.** 8. ...Final Answer: $\boxed{7}$

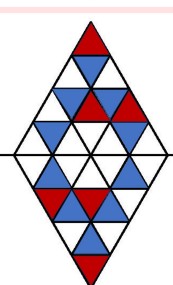

The model reasoning suffers from two key issues. First, it failed to "remember" prior pairings, which humans would track easily using the diagram, revealing its inability to manage dynamic states in multi-step tasks without visual feedback. Second, tasks without visual feedback. Second, it ignored image-derived constraints, like the "3-blue pairs" rule, and wrongly assumed pairing like colors was natural. This highlights the model's broader limitation in integrating textual and visual information, particularly for tasks requiring spatial reasoning and multi-hop logic, where visual aids simplify the solution.

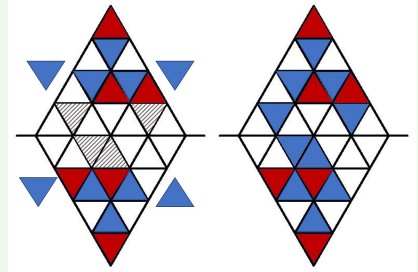

**Human Reasoning (Ground Truth):**
Forming the remaining 2 top B and 2 bottom B, the image shows there are already 3 B-B pairs. Thus, the 2 top B and 2 bottom B cannot form more B-B pairs but instead create 4 B-W pairs. After this, the remaining W-W pairs total 5, as counted in the image.

**Figure 21:** A sample error case in math.

## Math: 3D Spatial Simulation 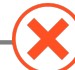

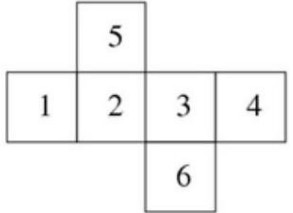

**Question:**

The diagram shows the net of a cube whose faces are numbered.

Sascha adds the numbers that are on opposite faces of the cube.

Which three results does he get?

**GPT-4o:**

To solve this problem, we need to determine which faces of the cube are opposite each other and then add the numbers on those opposite faces.

1. Step 1: Identify opposite faces: The diagram shows the net of a cube. In a cube, each face has an opposite face. From the net, we can deduce the following pairs of opposite faces: (1)Face 5 is opposite face 6. (2)Face 1 is opposite face 4. (3)Face 2 is opposite face 3.

2. Step 2: Add the numbers on opposite faces: Now, we add the numbers on the opposite faces: 5 + 6 = 11, 1 + 4 = 5, 2 + 3 = 5

3. Step 3: List the results: The three results Sascha gets are 11, 5, and 5.

Final Answer: \boxed{11, 5, 5}

**GPT-4o-caption:** Totally Correct

1. Main Structure: The image depicts the net of a cube, consisting of six numbered square faces. The net layout includes: A center square labeled 2. Three squares adjacent to the center square, forming an inverted T: 1 on the left, 3 on the right, 5 on the top. A square labeled 4, located to the right of 3, extending horizontally. A square labeled 6, located below 3, extending vertically.

2. Task: Sascha is tasked with folding the cube, determining the opposite faces, and adding the numbers on those faces.

**o1-preview:**

To solve this problem, we need to determine which faces of the cube are opposite each other after folding the given net, and then calculate the sums of the numbers on those opposite faces.

1. Step 1: Visualizing the Net and Cube Folding. Let's first reconstruct the net as described: Square 2 is in the center, Square 1 is to the left of Square 2, Square 3 is to the right of Square 2, Square 5 is above Square 2, Square 4 is to the right of Square 3, Square 6 is below Square 3.

```
        [5]
   [1] [2] [3] [4]
        [6]
```

2. Step 2: Assigning Squares to Cube Faces. We can assign the squares to the faces of the cube as follows: Front Face: Square 3, Back Face: Square 4 Left Face: Square 1, Right Face: Square 6, Top Face: Square 5, Bottom Face: Square 2

3. Step 3: Identifying Opposite Faces. The pairs of opposite faces are: (1). Front 3 and Back 4 (2). Left 1 and Right 6 (3). Top 5 and Bottom 2

4. Step 4: Calculating the Sums. (1). Front and Back Faces: [3 + 4 = 7] (2). Left and Right Faces: [1 + 6 = 7] (3). Top and Bottom Faces: [5 + 2 = 7]

Final Answer: \boxed{7, 7, 7}

**Error Reason:**

This image can be accurately captioned by GPT-4o, as the descriptive content is totally correct. However, after extracting the information, the model needs to visualize the process of folding the 2D shape to a 3D cube and determine the relative positions of each face. Unfortunately, both GPT-4o and GPT-o1 fail to perform the correct spatial reasoning required for this task.

**Ground Truth: 4, 6, 11**

**Figure 22:** A sample error case in math.

**Math: 2D Transformation** ❌

<image1>:

<image2>:

(A) (B) (C) (D) (E)

**Question:**

Anna cuts the picture of a mushroom in two halves. She then arranges the two pieces together to form a new picture. What could this new picture look like?

**GPT-4o:**

1. Understanding the Problem: In the first image (image1), Anna cuts the mushroom picture vertically into two halves. The task is to determine how the two halves can be rearranged to form a new picture.

2. Analyzing the Cut: The mushroom is cut vertically down the middle, so the left half of the mushroom (including half of the cap and half of the stem) is separated from the right half. The two halves can be rearranged in different ways, such as flipping or rotating them.

3. Analyzing the Options: Option A: The two halves are placed side by side, but they are not flipped or rotated. This looks like the original mushroom, just split in half. Option B: The two halves are placed side by side, but the right half is flipped horizontally. This creates a symmetrical image. Option C: The two halves are placed one on top of the other, but they are not flipped or rotated. This does not seem to form a logical image. Option D: The two halves are placed one on top of the other, but the top half is flipped horizontally. This also does not form a logical image. Option E: The two halves are placed one on top of the other, but the bottom half is flipped horizontally. This does not form a logical image either.

4. Conclusion: The only option that forms a new, symmetrical picture is Option B, where the right half is flipped horizontally and placed next to the left half. Final Answer: \boxed{B}

**GPT-4o-caption:**

1. Top Section <image1>: The image shows a drawing of a mushroom. The mushroom is symmetrical along a vertical axis. The mushroom has two distinct parts: Cap: A rounded, brown shape on top. Stem: A white oval base connected to the cap. A dashed vertical line divides the mushroom into two equal halves, splitting the cap and stem down the middle. At the bottom, there is a small drawing of scissors, suggesting that the mushroom will be physically cut into two vertical halves.

2. Bottom Section <image2>: Below the original mushroom, there are five rectangular images labeled (A), (B), (C), (D), and (E). Each rectangle depicts a possible arrangement of the two mushroom halves. (A): The two halves are placed side-by-side in their original orientation, forming a full mushroom when combined correctly. (B): The two halves are swapped and flipped, so the stem sections point outward while the cap sections meet at the center. (C): The two halves are rotated $90°$ clockwise, creating a horizontal arrangement with the cap on the left and the stem on the right. (D): The two halves are stacked vertically, with the cap on top and the stem below. (E): The two halves are flipped horizontally, creating a mirrored image where the cap sections face outward.

3. The task is to determine which of the five options correctly represents a new arrangement of the mushroom halves after being cut and rearranged.

**o1-preview:**

To solve this problem, let's analyze the possible arrangements of the two mushroom halves after cutting it vertically.

1. Given: The mushroom is symmetrical along a vertical axis. After cutting it vertically, we have a left half and a right half, each containing half of the cap and half of the stem.

2. Possible manipulations: (a) Flip a half horizontally (mirror image over the vertical axis). (b) Rotate a half (e.g., $90°$ clockwise). (c) Swap the halves.

3. Analyze options: Option A: Placing the halves together in their original orientation forms the same picture, not a new one. So, it's not acceptable as a new arrangement. Option B: Swapping and flipping both halves horizontally results in the cap sections meeting at the center and the stem sections pointing outward. This creates a new symmetrical image different from the original. Option C: Rotating both halves $90°$ clockwise and placing them side by side can reconstruct the mushroom horizontally. However, since the halves are mirror images, the cap and stem won't align properly to form a coherent image. Option D: Stacking the halves vertically doesn't form a meaningful new picture because both halves have parts of the cap and stem, and stacking them won't align these parts sensibly. Option E: Flipping both halves horizontally and placing them together will make the stem sections meet at the center and the cap sections face outward. This is a possible new arrangement.

4. Conclusion:

Both Option B and Option E are possible new arrangements. However, since we need to choose one, and Option B directly follows from swapping and flipping the halves as described, Option B is the best choice.

**Error Reason:**

When using GPT-4o to caption the image, it fails to accurately determine whether the image has been flipped, rotated or swapped. Human perform such reasoning by mentally simulating the transformation of the image, a capability we believe current MLLMs lack. This process involves visual reasoning, which cannot be achieved through text-based reasoning alone.

**Ground Truth: E**

**Figure 23:** A sample error case in math.

## Math: Pattern Inference 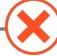

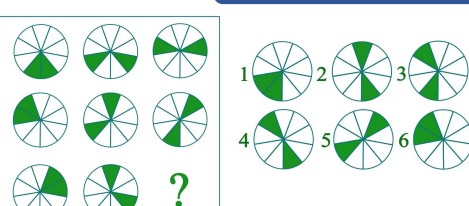

**Question:**

Choose the answer for the missing picture.

**GPT-4o:**

1. Rows and Columns: The grid is a 3x3 matrix, and each circle is divided into 8 equal sections, with some sections shaded green.

2. Row-wise Analysis: (A) First Row: The number of green sections increases by 1 as you move from left to right. First circle: 2 green sections. Second circle: 3 green sections. Third circle: 4 green sections. (B) Second Row: The number of green sections also increases by 1 as you move from left to right. First circle: 3 green sections. Second circle: 4 green sections. Third circle: 5 green sections. (C) Third Row: The number of green sections should follow the same pattern, increasing by 1 as you move from left to right. First circle: 4 green sections. Second circle: 5 green sections. Third circle (missing): It should have 6 green sections.

3. Conclusion: The missing circle should have 6 green sections. Now, let's look at the options: Option 6 has 6 green sections. Thus, the correct answer is 6.

**GPT-4o-caption:**

The image contains a pattern-based reasoning puzzle with the following key elements:

1. Main Puzzle Grid (Top Section): A 3x3 grid where the first eight cells are filled with circular diagrams, and the bottom-right cell is marked with a question mark ("?"). Each circular diagram is divided into eight equal sectors. Some sectors are shaded in green, while others remain white. The shaded green sectors vary in number and position, showing distinct patterns across rows and columns.

2. Rows in the Main Grid: Row 1: Cell 1: One green sector in the bottom-left position. Cell 2: Two adjacent green sectors in the left quadrant. Cell 3: Three adjacent green sectors, starting from the left and moving clockwise. Row 2: Cell 1: Two green sectors, one in the top-right and one in the bottom-right quadrant. Cell 2: Three adjacent green sectors, forming the right half of the circle. Cell 3: Four adjacent green sectors, forming a "V" shape covering the bottom and right quadrants. Row 3: Cell 1: Three green sectors, forming a pattern in the bottom half. Cell 2: Four adjacent green sectors, creating an "L" shape in the left half. Cell 3: Missing circle (marked by "?").

3. Answer Options (Bottom Section): Below the main grid, six circular diagrams labeled 1 to 6 are presented as potential answers. Each diagram follows the same structure (eight equal sectors, some shaded green). The options are as follows: Option 1: Three green sectors in the bottom-left and bottom-center quadrants. Option 2: Four adjacent green sectors covering the top and right quadrants. Option 3: Three green sectors in the top half, symmetric across the vertical axis. Option 4: Four adjacent green sectors, covering the bottom-left and bottom quadrants. Option 5: Four green sectors forming a cross, symmetric across both axes. Option 6: Five adjacent green sectors forming a large section from the bottom-left to the top-right quadrant.

**o1-preview:**

1. Understand the Grid and the Circles: Each circle is divided into 8 equal sectors. Some sectors are shaded green, and others are white. The shading patterns vary across the grid. (A) Grid Details: (a) Row 1: Cell (1,1): 1 green sector (bottom-left position). Cell (1,2): 2 adjacent green sectors in the left quadrant. Cell (1,3): 3 adjacent green sectors, starting from the left and moving clockwise. (b) Row 2: Cell (2,1): 2 green sectors in the top-right and bottom-right positions. Cell (2,2): 3 adjacent green sectors forming the right half of the circle. Cell (2,3): 4 adjacent green sectors forming a "V" shape covering the bottom and right quadrants. (c) Row 3: Cell (3,1): 3 green sectors forming a pattern in the bottom half. Cell (3,2): 4 adjacent green sectors creating an "L" shape in the left half. Cell (3,3): Missing circle (marked by "?").

2. Identify Patterns in the Grid: (a) Number of Green Sectors: In each row, the number of green sectors increases by 1 from left to right. In each column, the number of green sectors increases by 1 from top to bottom. (b) Logical Deduction: The missing circle at Cell (3,3) should have 5 green sectors to continue the pattern. (c) Shading Pattern Direction: The shading seems to progress in a particular direction in each row.

3. Predict the Missing Circle's Shading Pattern: Since Cell (3,2) has 4 adjacent green sectors forming an "L" shape in the left half, it's logical to predict that Cell (3,3) will have 5 adjacent green sectors continuing from where the previous shading left off. The 5 green sectors likely form a contiguous area, possibly starting from the bottom-left sector and covering up to the top-right sector.

4. Match the Prediction with the Answer Options: Option 6 shows a circle with 5 adjacent green sectors forming a large section from the bottom-left to the top-right quadrant, which matches our prediction.

5. Conclusion: Based on the logical patterns observed in the grid, Option 6 is the correct answer.

**Error Reason:**

Both GPT-4o and GPT-o1 incorrectly identify the pattern: they conclude that the number of green sections changes in a regular pattern. However, it's easy to find the actual rule is that the position of the green sections changes in a regular pattern. By misidentifying the color, quantity and position, both models perform flawed reasoning and generate wrong answer.

**Ground Truth: 4**

**Figure 24:** A sample error case in math.

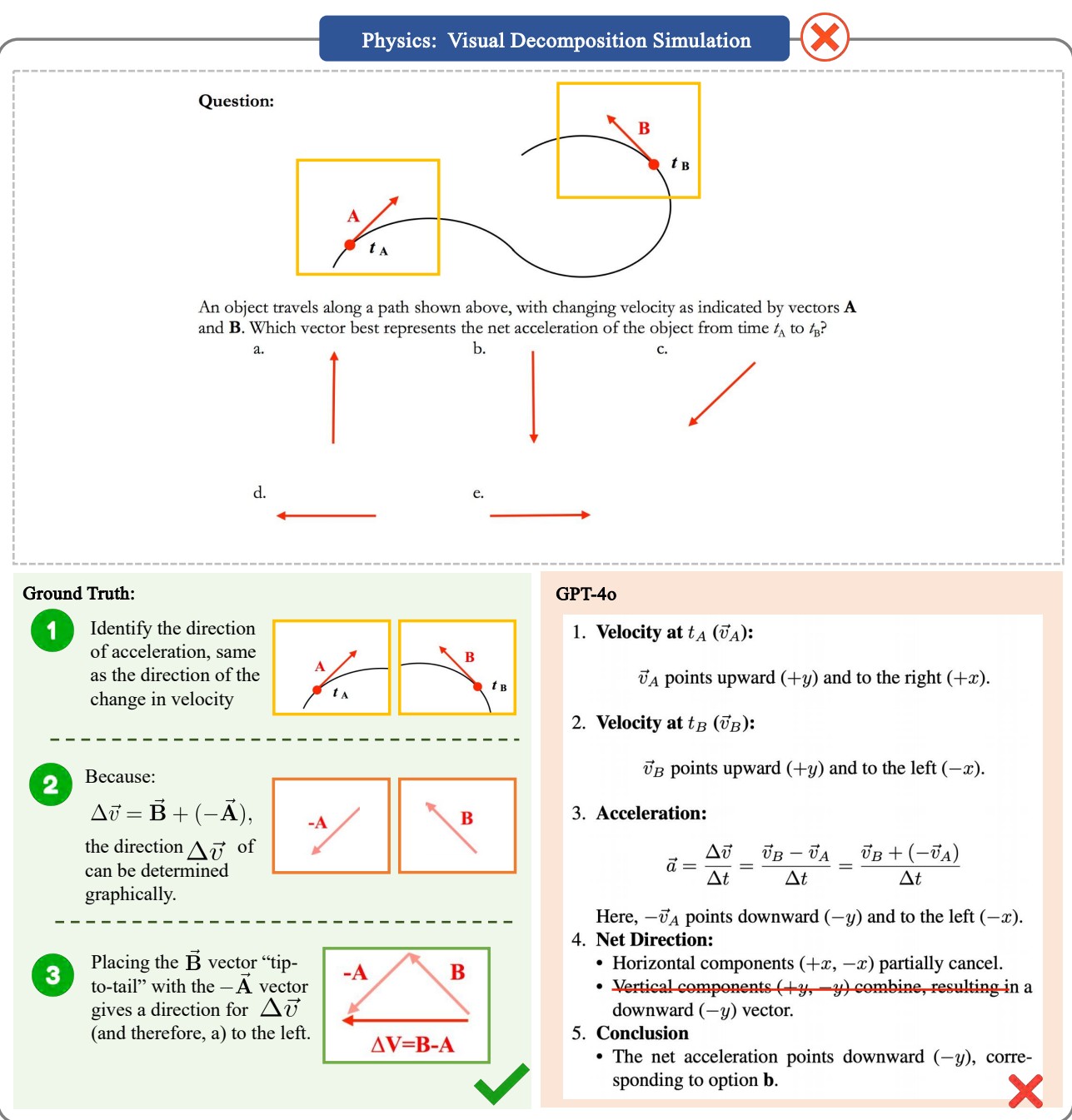

**Figure 25:** A sample error case in physics.

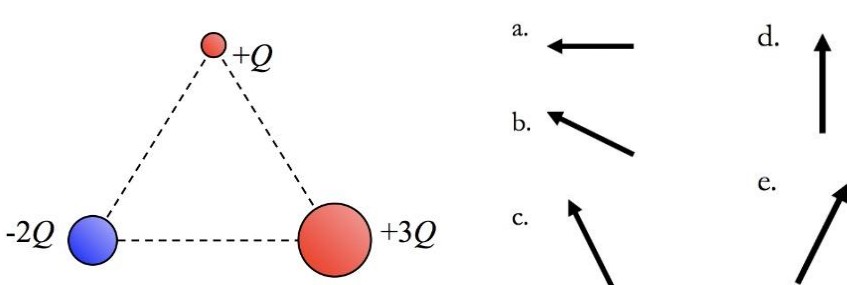

## Physics: 3D Field Simulation 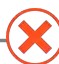

**Question:** Three point charges, of charge **+Q**, **-2Q**, and **+3Q**, are placed equidistant as shown. Which vector best describes the net direction of the electric force acting on the **+Q** charge?

### GPT-4o with CoT :

1. **Forces Due to the Charges**:
   - The $-2Q$ charge attracts the $+Q$ charge. (Force points downward-left).

   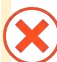 • The $+3Q$ charge repels the $+Q$ charge. (Force points downward-right).

2. **Force Magnitudes (Coulomb's Law)**:
   - Force due to $-2Q$: $F_{-2Q} = \frac{2kQ^2}{r^2}$.
   - Force due to $+3Q$: $F_{+3Q} = \frac{3kQ^2}{r^2}$.

3. **Force Directions**:
   - $-2Q$: Downward-left.
   - $+3Q$: Downward-right.

4. **Force Components** (at $60°$):
   - Horizontal:
     $F_{-2Q,x} = -\frac{kQ^2}{r^2}, F_{+3Q,x} = +\frac{3kQ^2}{2r^2}$.

   - Vertical:
     $F_{-2Q,y} = -\frac{\sqrt{3}kQ^2}{r^2}, F_{+3Q,y} = -\frac{3\sqrt{3}kQ^2}{2r^2}$

5. **Net Force Components**:
   - Horizontal: $F_x = \frac{kQ^2}{2r^2}$.
   - Vertical: $F_y = -\frac{5\sqrt{3}kQ^2}{2r^2}$.

6. **Net Force Direction**:

   The net force is determined by

   $$\tan^{-1}\left(\frac{F_y}{F_x}\right).$$

   $F_x > 0$ and $F_y < 0$ indicate the direction is downward-right.

   **Final Answer:** $\boxed{e}$.

**Direct GPT-4o : b.** ✅

**Error Analysis:** Misunderstand the net force's direction during vector addition. The issue arose from not correctly identifying the force direction from **+3Q** to **+Q** (longer red arrow in vector graphic on the right), which causes the net force to point ↘, not downward-right ↘ in ⊗ .

**Human Reasoning (Ground Truth):**
The direction of the electric force due to **-2Q** and **+3Q** can be visualized with a

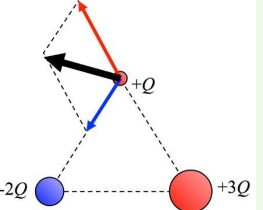 quick free-body sketch of the electric forces:

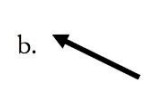

**Figure 26:** A sample error case in physics.

**Physics: Visual Decomposition Simulation**

<image_1>

View looking down on pool table

a. b. c. d. e.

<image_2>

**Question:**

A billiard ball hits the side of a pool table at an angle Ø as shown in the top view above, and bounces away at the same angle, and with the same speed. Which vector indicates the direction of the net change in momentum of the billiard ball?

**GPT-4o:**

**To determine the direction of the net change in momentum of the billiard ball:**

1. **Initial Momentum:**
   The ball is moving towards the wall at an angle Θ to the vertical.

2. **Final Momentum:**
   After bouncing off the wall, the ball moves away at the same angle Θ, but the horizontal direction reverses.

Since the speed of the ball remains constant, the magnitude of the momentum does not change, only the direction does.

**Analyzing the Components of Momentum:**

- The **horizontal component** of the momentum changes direction (e.g., from left to right).
- The **vertical component** remains unchanged.

This results in a net change in momentum directed horizontally to the **right**.

**Conclusion:**

The vector representing the net change in momentum points **to the right**.

The correct answer is **(b)**.

**Error Analysis:**

The answer incorrectly assumes that the momentum in the y direction does not change, thus drawing the wrong conclusion: the net change of momentum is in the horizontal direction.

**Ground Truth:**

The correct answer is *d*. The billiard ball has velocity vectors in both the *x* and *y* directions, as shown here. The *x*-component of the velocity remains unchanged, but the *y*-component has changed from the positive to the negative direction, a change in velocity—and thus, momentum—in the negative *y* direction.

$v_{x\text{-}initial}$ $v_{x\text{-}final}$

$v_{y\text{-}initial}$ $v_{y\text{-}final}$

**Figure 27:** A sample error case in physics.

## Physics: Graph Reasoning 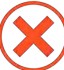

**Question:**

An object is dropped and accelerates downwards. As it falls it is affected by air friction, but never reaches terminal velocity during the course of its fall. The graph that could indicate the magnitude of the object's acceleration as a function of time is

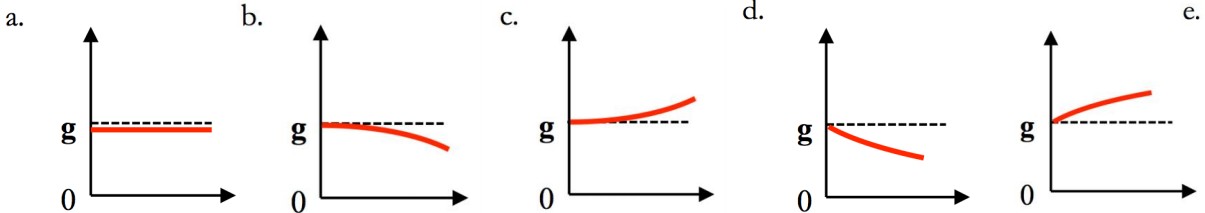

**GPT-4o:**

When an object is dropped, it initially accelerates at the acceleration due to gravity, g . As it falls, air resistance (friction) increases, which reduces the net acceleration. However, since the object never reaches terminal velocity, the acceleration decreases but does not reach zero. Let's analyze the options:

a. The graph shows constant acceleration at g , which would be the case if there were no air resistance. This is not correct because the object is affected by air friction.

b. The graph shows acceleration starting at g and gradually decreasing, which is consistent with the object being affected by air resistance. The acceleration decreases over time but does not reach zero, which matches the scenario where the object does not reach terminal velocity.

c. The graph shows acceleration increasing over time, which is incorrect because air resistance would cause the acceleration to decrease, not increase.

d. The graph shows acceleration starting at g and decreasing rapidly, which could represent a situation where the object reaches terminal velocity. However, the problem states that the object does not reach terminal velocity, so this is not the correct option.

e. The graph shows acceleration starting below g and increasing, which is incorrect because the object starts with an acceleration of g and air resistance causes it to decrease.

The answer is b.

**GPT-4o Caption:**

The image shows five graphs labeled a through e, each representing acceleration versus time:
a. The graph is a horizontal line at a constant value of g .
b. The graph starts at g and curves downward, decreasing over time.
c. The graph starts below g and curves upward, increasing over time.
d. The graph starts at g and curves downward, decreasing over time.
e. The graph starts below g and curves upward, increasing over time.
Each graph has a dashed line indicating g as a reference point.

**Figure 28:** A sample error case in physics.

## Physics: Graph Reasoning

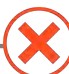

### o1-preview:

To determine which graph correctly represents the magnitude of the object's acceleration as a function of time, let's analyze the physics of the situation step by step.

**Understanding the Physical Scenario**

- **Object in Free Fall with Air Resistance:**
  - When an object is dropped, it initially accelerates downward due to gravity with acceleration $g$.
  - As it gains speed, air resistance (a type of friction) opposes the motion.
  - Air resistance increases with velocity, reducing the net acceleration.
  - The object **never reaches terminal velocity** during its fall, so it continues to accelerate but at a decreasing rate.

**Mathematical Analysis**

1. **Net Force on the Object:**

   The net force $F_{\text{net}}$ acting on the object is:

   $$F_{\text{net}} = mg - kv$$

   where:
   - $m$ is the mass of the object.
   - $g$ is the acceleration due to gravity.
   - $k$ is the air resistance constant.
   - $v$ is the velocity of the object at time $t$.

2. **Newton's Second Law:**

   Applying Newton's second law $F = ma$:

   $$ma = mg - kv$$

   $$a = g - \frac{k}{m}v$$

   This shows that acceleration $a$ decreases as velocity $v$ increases.

3. **Velocity as a Function of Time:**

   Solving the differential equation:

   $$\frac{dv}{dt} = g - \frac{k}{m}v$$

   The solution is:

   $$v(t) = \frac{mg}{k}\left(1 - e^{-\frac{k}{m}t}\right)$$

   where $v(0) = 0$ (since the object is dropped from rest).

4. **Acceleration as a Function of Time:**

   Substituting $v(t)$ back into $a(t)$:

   $$a(t) = g - \frac{k}{m}v(t) = ge^{-\frac{k}{m}t}$$

   This shows that acceleration decreases exponentially over time from $g$ towards zero.

### o1-preview:

**Choosing the Correct Graph**

Given the above analysis, the graph of acceleration versus time should:

- **Start at $g$:** Because the initial acceleration is $g$ when $t = 0$.
- **Decrease Exponentially:** The acceleration decreases exponentially due to the increasing effect of air resistance.
- **Never Reach Zero:** Since the object never reaches terminal velocity during its fall, the acceleration approaches zero but never quite reaches it within the time frame considered.

**Graph Descriptions:**

- **Option (a):** A horizontal line at $g$ (constant acceleration). **Incorrect** because it doesn't account for decreasing acceleration due to air resistance.
- **Option (b):** Starts at $g$ and curves downward, decreasing over time. **Correct** as it represents exponential decay of acceleration.
- **Option (c):** Starts below $g$ and curves upward, increasing over time. **Incorrect** because acceleration doesn't increase in this scenario.
- **Option (d):** Starts at $g$ and curves downward, decreasing over time. Since options (b) and (d) have the same description, but assuming one represents linear decrease and the other exponential, (d) may represent a linear decrease, which is **incorrect**.
- **Option (e):** Starts below $g$ and curves upward, increasing over time. **Incorrect** for the same reason as (c).

**Conclusion**

The graph that correctly represents the magnitude of the object's acceleration as a function of time is:

Option (b):

- **Starts at $g$ and decreases exponentially over time.**

### Ground Truth: (d)

### Error Analysis:

Misunderstanding the characteristics of exponential decay, it is believed that the declining trend of option B is reasonable, but overlooks that the decline of D is more gradual and better fits real physical phenomena. Moreover, the description in the question that "the acceleration will never reach 0" cannot be correctly matched to the image of any option.

**Figure 29:** A sample error case in physics.

## Physics: 3D Field Simulation ✅

<image_1>

**Question:** Two long parallel wires, oriented along the $y$-axis as shown, contain currents $I$ and $2I$, flowing in opposite directions. Which of the following statements is true?
  a. The magnetic field due to $I$ circles that wire in a counterclockwise direction.
  b. The magnetic force from wire $2I$ on wire $I$ is twice as strong as the force on wire $I$ from wire $2I$.
  c. The magnetic force between the two wires is proportional to the inverse square of the distance between them.
  d. The force on wire $2I$ is in the $+x$ direction.
  e. The force on wire $2I$ is in the $-z$ direction.

### GPT-4o:

**Analysis:**

1. **Magnetic Field Direction (Right-Hand Rule):**
   - Current $I$ (into the page): magnetic field circles clockwise.
   - Current $2I$ (out of the page): magnetic field circles counterclockwise.
   **Option a** is incorrect because $I$'s field is clockwise, not counterclockwise.

2. **Magnetic Force Between Wires:**
   - By Ampère's law, force per unit length is proportional to $I_1 I_2/d$.
   - Forces are equal in magnitude but opposite in direction (Newton's third law).
   **Option b** is incorrect.

3. **Dependence on Distance:**
   - Force is inversely proportional to $d$, not $d^2$.
   **Option c** is incorrect.

4. **Force Direction:**
   - Opposite currents repel. The force on $2I$ points in $+x$-direction.
   **Option d** is correct; **Option e** is incorrect.

**Conclusion:**

The answer is **d**.

### Ground Truth:

The correct answer is $d$. The magnetic field due to wire $I$ circles that wire in a clockwise direction, as determined using the Right Hand Rule. This field is down (in the $-z$ direction) at the location of wire $2I$. By applying another Right Hand Rule, we can determine that the magnetic force on $2I$ due to this magnetic field is in the $+x$ direction.

Wire $2I$ has a force equal in magnitude on wire $I$. The magnetic force between the two wires can be determined as follows:
  Magnitude of magnetic field due to $I$:

$$B = \frac{\mu_0 I}{2\pi r}, \text{ where } r \text{ is the distance between the wires}$$

Force on wire $2I$ due to this field:

$$F_B = I\ell \times B$$

$$F_B = (2I)\ell \times \frac{\mu_0 I}{2\pi r}$$

The force per unit length of the wires, then, is:

$$\frac{F_B}{\ell} = \frac{\mu_0 2I^2}{2\pi r}$$

**Figure 30:** A sample correct case in physics.

## Physics: Graph Reasoning 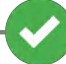

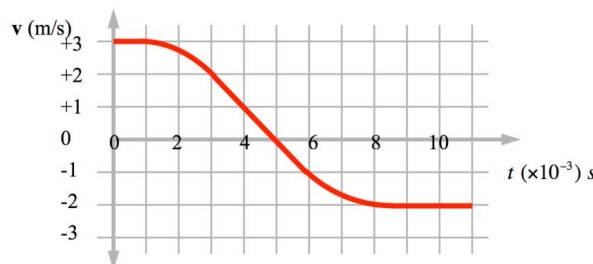

**Question:**

A billiard ball rolling across a table in the +x direction at 3 m/s hits the edge of the table at aperpendicular angle, and bounces back in the -x direction, now traveling at 2 m/s in theopposite direction, 'The greatest magnitude of acceleration for the billiard ball was at time

a. $t = 1 \times 10^{-3} s$   b. $t = 2 \times 10^{-3} s$

c. $t = 5 \times 10^{-3} s$   d. $t = 7 \times 10^{-3} s$   e. $t = 9 \times 10^{-3} s$

**GPT-4o:**

The graph shows the velocity of the billiard ball as a function of time. Initially, the ball is moving in the +x direction at 3 m/s, and then it slows down, reverses direction, and moves in the -x direction at 2 m/s .

The greatest magnitude of acceleration occurs when the velocity changes most rapidly, which corresponds to the steepest part of the velocity-time graph. From the graph, the steepest slope occurs around $t = 5 \times 10^{-3} s$, where the ball is reversing direction.

The answer is c.

**Ground Truth:**

The correct answer is *c.* Acceleration is defined as the change in velocity over time:

$$a = \frac{\Delta v}{\Delta t} = \frac{v_{final} - v_{initial}}{t_{final} - t_{initial}}$$

Because *v* here is plotted on the *y*-axis and *t* is on the *x*-axis, $\frac{\Delta v}{\Delta t} = \frac{rise}{run} = m$ , the slope of the graph. The slope of a velocity-time graph represents the acceleration of the object, so the greatest acceleration occured where the slope was greatest. Here, that would be at any time between *t* = 3 and *t* = 6 milliseconds.

**Figure 31:** A sample correct case in physics.

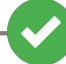

## Chemistry: Reaction Simulation

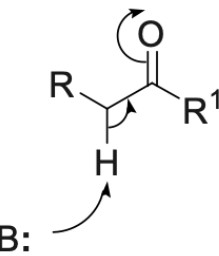

**Question:**

An 'arrow-pushing' diagram is a common type of chemical image used to illustrate electron flow in mechanistic steps. Choose the molecular image that represents the SMILES expression for the molecule after the electron has been relocated, as depicted in the image.

A. C=C(/C[O-])C(C)=O

B. C=C(/C(C)=O)[O-]

C. C=C(/O)C(C[O-])=O

D. */C=C(/*)[O-].*C(*)=O

**GPT-4o:**

Let's analyze the arrow-pushing mechanism step by step:

1. The image shows a carbonyl group (C=O) with an arrow indicating electron movement from the double bond to the oxygen atom. This suggests that the oxygen will gain a negative charge, becoming an alkoxide ion (O⁻).

2. The second arrow shows the bond between the hydrogen (H) and the adjacent carbon breaking, with the electrons moving towards the carbon-carbon bond. This suggests the formation of a double bond between the two carbon atoms.

3. The result of this electron movement is the formation of a new C=C double bond and the oxygen becoming negatively charged (O⁻).

Now, let's examine the options:

- Option A: This structure shows a C=C double bond, a negatively charged oxygen (O⁻) attached to a carbon, and a carbonyl group (C=O). This does not match the expected structure because the negative charge is on the wrong oxygen.

- Option B: This structure shows a C=C double bond, a negatively charged oxygen (O⁻), and a carbonyl group (C=O) attached to the same carbon. This matches the expected structure after the electron movement.

- Option C: This structure shows a C=C double bond, but the negative charge is on a different oxygen, which does not match the expected structure.

- Option D: This structure is incomplete and does not match the expected structure.

**The answer is \boxed{B}.**

**Ground Truth: B**

**Figure 32:** A sample correct case in chemistry.

## Chemistry: Knowledge-based Counting 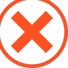

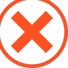

**Question:** In the transition-state structure shown in the image, calculate the total number of bonds in the structure, including single, double, and triple bonds but excluding those involving hydrogen. Note: Disregard arrows. Consider all components present in the transition-state structure shown in the image.

<image_1>

**o1:** A clear way to see why there are 17 total bonds (excluding those to hydrogen) is to break the structure into three parts and count them: 1. Cyclohexyl moiety: Six C–C single bonds within the cyclohexane ring. 2. Tetrahedral region around the acyl center: One single bond between the oxygen and the carbonyl carbon. One single bond between the same oxygen and the cyclohexyl carbon. One double bond in the acyl group (C=O). One single bond between the carbonyl carbon and the aromatic ring. 3. Aromatic ring with a chloro substituent: In the Kekulé representation, the benzene ring has three single bonds and three double bonds (six ring bonds total). One single bond between a ring carbon and the chlorine atom. Adding these together, and ignoring any bonds to hydrogen: Single bonds: 6 (cyclohexane ring) + 1 (O–C in tetrahedral region) + 1 (O–cyclohexyl carbon) + 1 (carbonyl C–aromatic ring) + 3 (single bonds in the benzene ring) + 1 (benzene–Cl) = 13 single bonds. Double bonds: 1 (C=O in the acyl group) + 3 (double bonds in the benzene ring) = 4 double bonds. Thus, the total is 13 + 4 = 17 bonds (excluding any to hydrogen).

**Ground Truth:** By directly counting the bonds marked on the diagram, we can see there are a total of **21 bonds**.

**Error Analysis:**

The error lies in undercounting the total number of bonds. The original answer missed four key single bonds: two **C–O bonds** within the ester group, **one O–C bond** connecting the ester oxygen to the cyclohexyl ring, and **one C–O bond** in the acyl center. These omissions led to a total of 17 bonds instead of the correct 21 bonds.

**Figure 33:** A sample error case in chemistry.

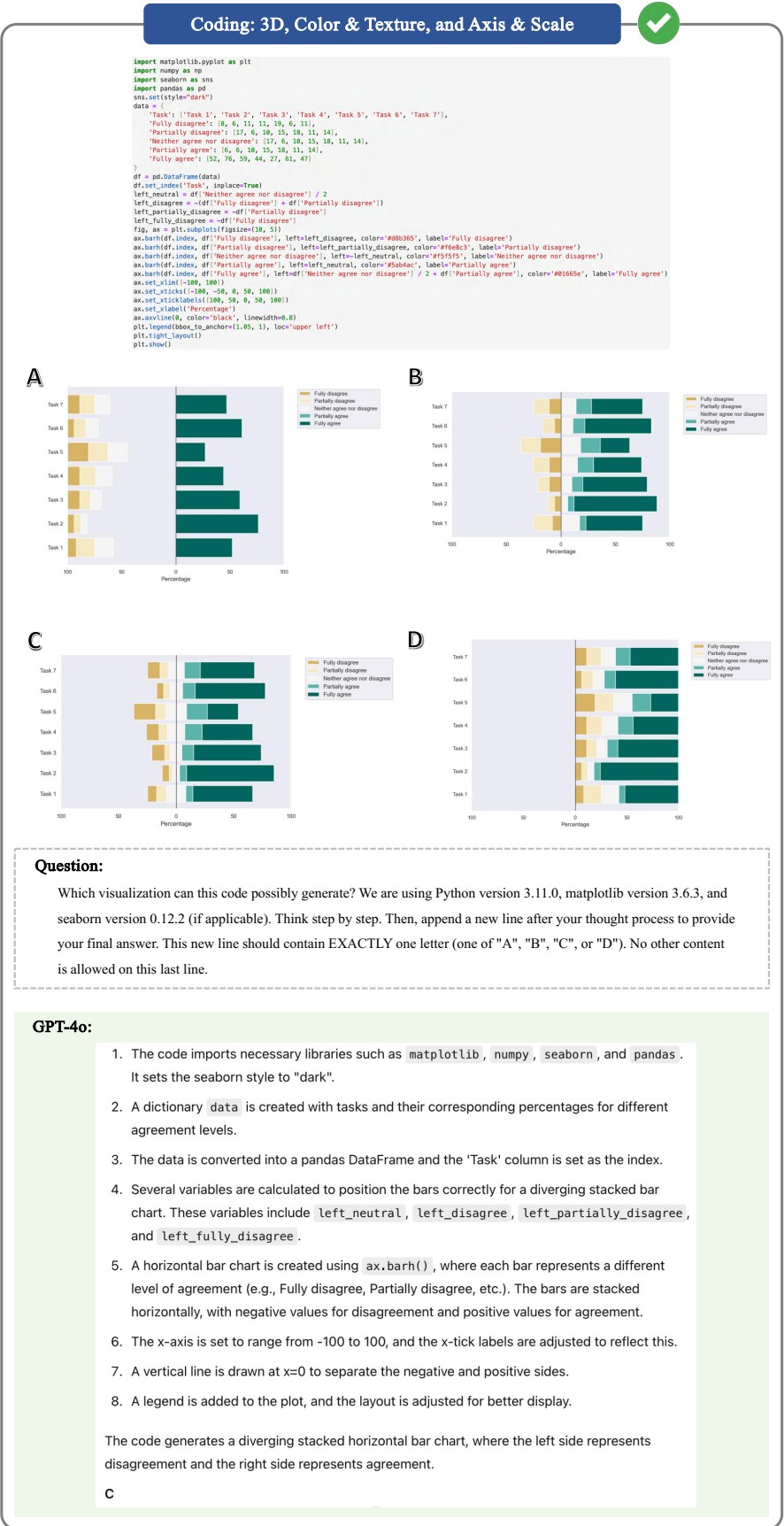

**Figure 34:** A sample correct case in coding.

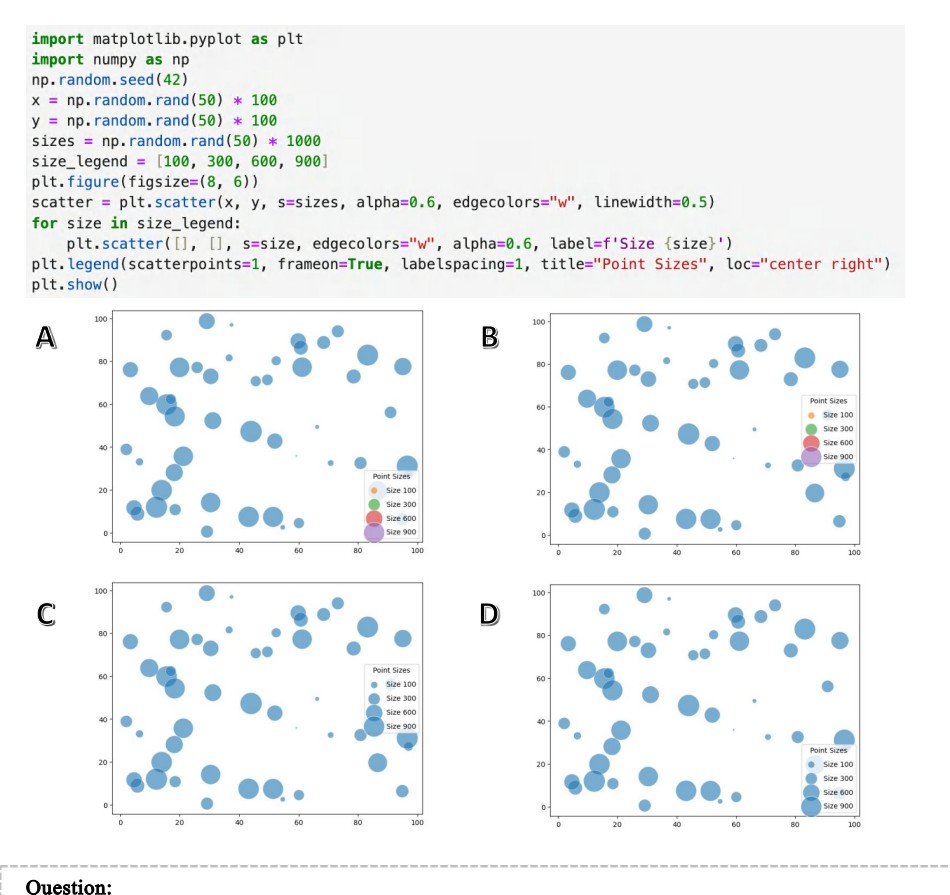

**Question:**

Which visualization can this code possibly generate? We are using Python version 3.11.0, matplotlib version 3.6.3, and seaborn version 0.12.2 (if applicable). Think step by step. Then, append a new line after your thought process to provide your final answer. This new line should contain EXACTLY one letter (one of "A", "B", "C", or "D"). No other content is allowed on this last line.

**GPT-4o:**

1. The code uses `np.random.rand(50)` to generate 50 random values for both `x` and `y`, which are scaled by 100. This means the scatter plot will have points with `x` and `y` values between 0 and 100.

2. The `sizes` array is generated using `np.random.rand(50) * 1000`, meaning the size of each point will vary between 0 and 1000.

3. The `plt.scatter()` function is used to create a scatter plot with the `x`, `y`, and `sizes` arrays. The `alpha=0.6` makes the points semi-transparent, and `edgecolors="w"` adds white edges to the points.

4. The `size_legend` list contains specific sizes `[100, 300, 600, 900]` for the legend. The code creates empty scatter points with these sizes to display in the legend.

5. The legend is placed at the "center right" of the plot, with the title "Point Sizes."

The key difference between the options is the color of the legend markers. Since the code does not specify any color for the legend markers, they will inherit the default color of the scatter plot, which is blue.

Final answer:

C

**Error Analysis:**

GPT-4o assumes that, since the code does not explicitly specify a color for the legend markers, they will inherit the default color of the scatter plot. However, it fails to account for the fact that the plt.scatter() function is called multiple times to render the markers in the legend. As a result, each call assigns a distinct color to the markers, meaning the final legend markers will not all be blue.

**Ground Truth:**

The code specifies that the legend is placed at the center right, which eliminates A and D. Since the plt.scatter() function is called multiple times (in a loop) to render the markers in the legend, each call assigns a distinct color to the markers. Therefore, the markers will assume different colors. **Correct Answer: B**

**Figure 35:** A sample error case in coding.

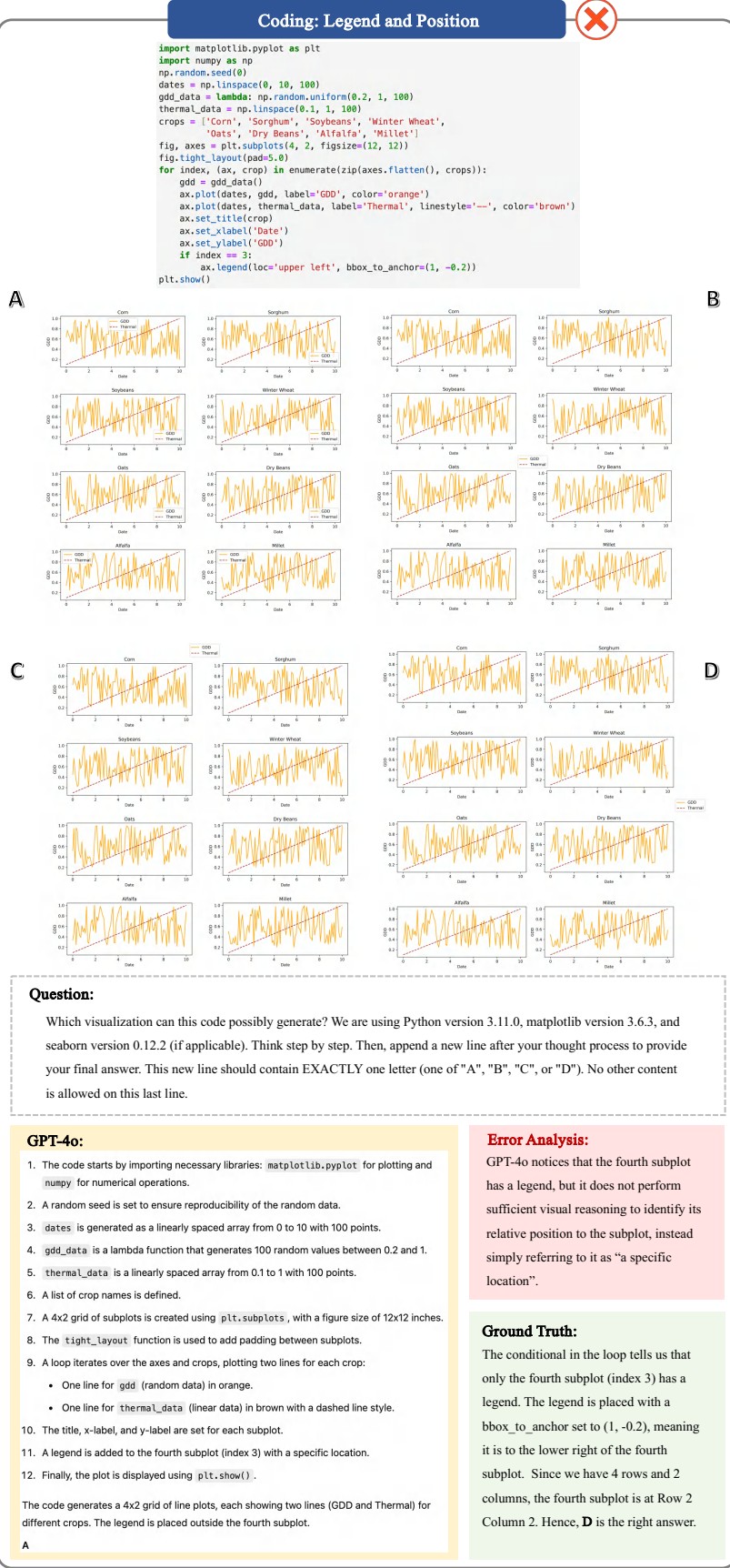

**Question:**

Which visualization can this code possibly generate? We are using Python version 3.11.0, matplotlib version 3.6.3, and seaborn version 0.12.2 (if applicable). Think step by step. Then, append a new line after your thought process to provide your final answer. This new line should contain EXACTLY one letter (one of "A", "B", "C", or "D"). No other content is allowed on this last line.

**GPT-4o:**

1. The code starts by importing necessary libraries: `matplotlib.pyplot` for plotting and `numpy` for numerical operations.

2. A random seed is set to ensure reproducibility of the random data.

3. `dates` is generated as a linearly spaced array from 0 to 10 with 100 points.

4. `gdd_data` is a lambda function that generates 100 random values between 0.2 and 1.

5. `thermal_data` is a linearly spaced array from 0.1 to 1 with 100 points.

6. A list of crop names is defined.

7. A 4x2 grid of subplots is created using `plt.subplots`, with a figure size of 12x12 inches.

8. The `tight_layout` function is used to add padding between subplots.

9. A loop iterates over the axes and crops, plotting two lines for each crop:
   - One line for `gdd` (random data) in orange.
   - One line for `thermal_data` (linear data) in brown with a dashed line style.

10. The title, x-label, and y-label are set for each subplot.

11. A legend is added to the fourth subplot (index 3) with a specific location.

12. Finally, the plot is displayed using `plt.show()`.

The code generates a 4x2 grid of line plots, each showing two lines (GDD and Thermal) for different crops. The legend is placed outside the fourth subplot.

A

**Error Analysis:**

GPT-4o notices that the fourth subplot has a legend, but it does not perform sufficient visual reasoning to identify its relative position to the subplot, instead simply referring to it as "a specific location".

**Ground Truth:**

The conditional in the loop tells us that only the fourth subplot (index 3) has a legend. The legend is placed with a bbox_to_anchor set to (1, -0.2), meaning it is to the lower right of the fourth subplot. Since we have 4 rows and 2 columns, the fourth subplot is at Row 2 Column 2. Hence, **D** is the right answer.

**Figure 36:** A sample error case in coding.

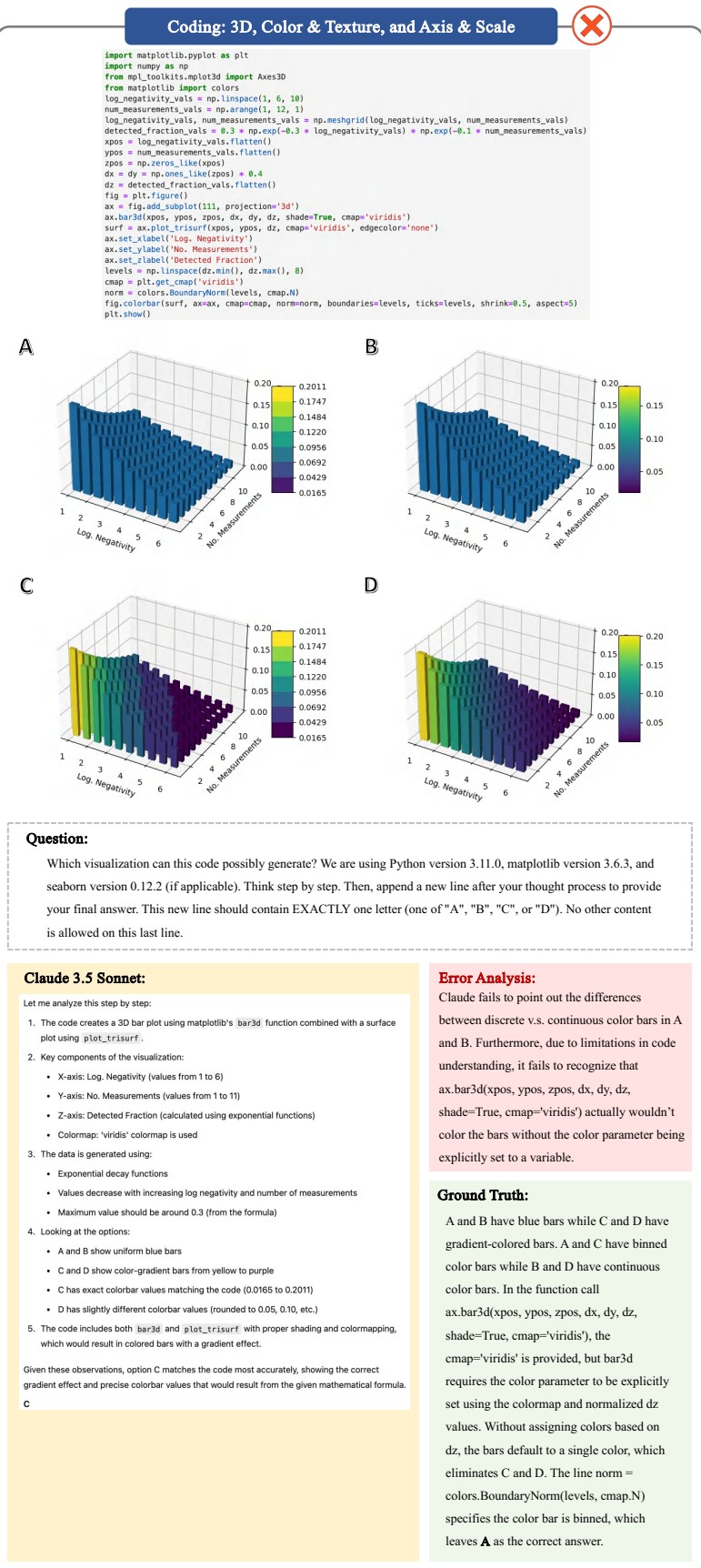

**Figure 37:** A sample error case in coding.

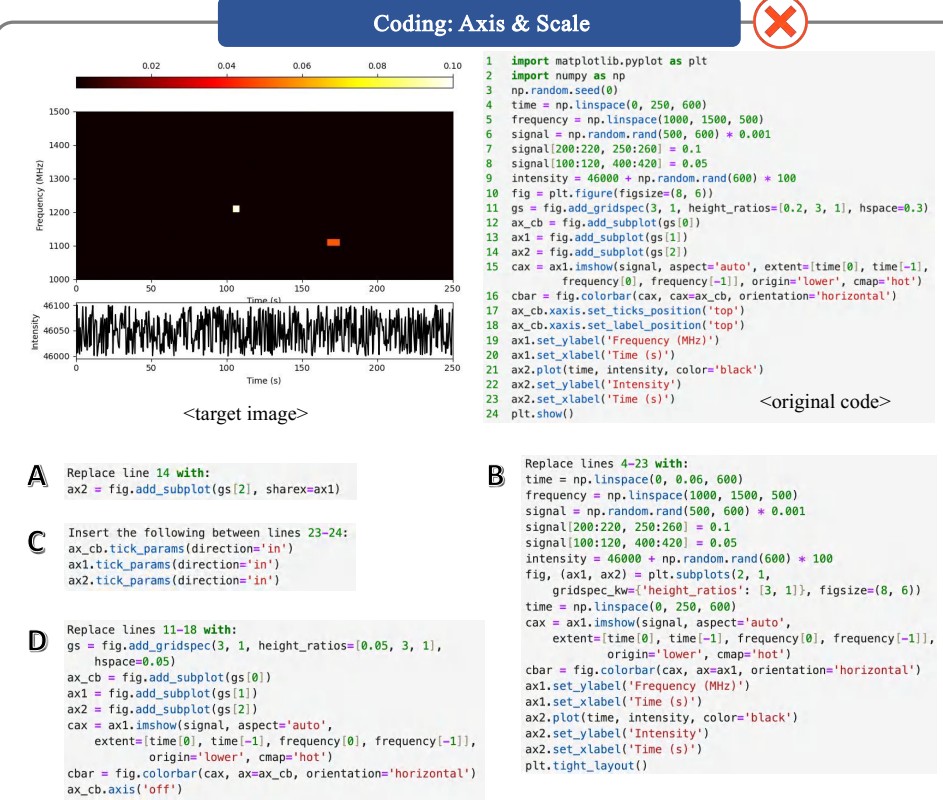

**Coding: Axis & Scale**

<target image>

<original code>
```python
import matplotlib.pyplot as plt
import numpy as np
np.random.seed(0)
time = np.linspace(0, 250, 600)
frequency = np.linspace(1000, 1500, 500)
signal = np.random.rand(500, 600) * 0.001
signal[200:220, 250:260] = 0.1
signal[100:120, 400:420] = 0.05
intensity = 46000 + np.random.rand(600) * 100
fig = plt.figure(figsize=(8, 6))
gs = fig.add_gridspec(3, 1, height_ratios=[0.2, 3, 1], hspace=0.3)
ax_cb = fig.add_subplot(gs[0])
ax1 = fig.add_subplot(gs[1])
ax2 = fig.add_subplot(gs[2])
cax = ax1.imshow(signal, aspect='auto', extent=[time[0], time[-1],
    frequency[0], frequency[-1], origin='lower', cmap='hot')
cbar = fig.colorbar(cax, cax=ax_cb, orientation='horizontal')
ax_cb.xaxis.set_ticks_position('top')
ax_cb.xaxis.set_label_position('top')
ax1.set_ylabel('Frequency (MHz)')
ax1.set_xlabel('Time (s)')
ax2.plot(time, intensity, color='black')
ax2.set_ylabel('Intensity')
ax2.set_xlabel('Time (s)')
plt.show()
```

**A** Replace line 14 with:
```python
ax2 = fig.add_subplot(gs[2], sharex=ax1)
```

**C** Insert the following between lines 23-24:
```python
ax_cb.tick_params(direction='in')
ax1.tick_params(direction='in')
ax2.tick_params(direction='in')
```

**D** Replace lines 11-18 with:
```python
gs = fig.add_gridspec(3, 1, height_ratios=[0.05, 3, 1],
    hspace=0.05)
ax_cb = fig.add_subplot(gs[0])
ax1 = fig.add_subplot(gs[1])
ax2 = fig.add_subplot(gs[2])
cax = ax1.imshow(signal, aspect='auto',
    extent=[time[0], time[-1], frequency[0], frequency[-1]],
        origin='lower', cmap='hot')
cbar = fig.colorbar(cax, cax=ax_cb, orientation='horizontal')
ax_cb.axis('off')
```

**B** Replace lines 4-23 with:
```python
time = np.linspace(0, 0.06, 600)
frequency = np.linspace(1000, 1500, 500)
signal = np.random.rand(500, 600) * 0.001
signal[200:220, 250:260] = 0.1
signal[100:120, 400:420] = 0.05
intensity = 46000 + np.random.rand(600) * 100
fig, (ax1, ax2) = plt.subplots(2, 1,
    gridspec_kw={'height_ratios': [3, 1]}, figsize=(8, 6))
time = np.linspace(0, 250, 600)
cax = ax1.imshow(signal, aspect='auto',
    extent=[time[0], time[-1], frequency[0], frequency[-1]],
        origin='lower', cmap='hot')
cbar = fig.colorbar(cax, ax=ax1, orientation='horizontal')
ax1.set_ylabel('Frequency (MHz)')
ax1.set_xlabel('Time (s)')
ax2.plot(time, intensity, color='black')
ax2.set_ylabel('Intensity')
ax2.set_xlabel('Time (s)')
plt.tight_layout()
```

**Question:**

Our goal is to reproduce the visualization in the image shown. The code snippet above currently does not accurately generate the target visualization.

We are using Python version 3.11.0, matplotlib version 3.6.3, and seaborn version 0.12.2 (if applicable). What change should we apply to the original code in order to generate the target visualization? Think step by step. Then, append a new line after your thought process to provide your final answer. This new line should contain EXACTLY one letter (one of "A", "B", "C", or "D"). No other content is allowed on this last line.

**GPT-4o:**

Let's analyze the target visualization and the provided code:

1. **Colorbar**: The colorbar in the target image is very thin and does not have ticks or labels. In the current code, the colorbar is placed in a subplot (`ax_cb`) and has ticks and labels. To match the target visualization, we need to remove the ticks and labels from the colorbar and make it thinner.

2. **Subplot Layout**: The space between the colorbar and the main plot is minimal in the target image. The current code uses a `height_ratios` of `[0.2, 3, 1]`, which gives the colorbar more space than necessary. We need to reduce the height of the colorbar subplot.

3. **Ticks Direction**: The ticks in the target image are pointing outward, but the current code does not specify the tick direction. This is a minor detail, but it can be adjusted.

4. **Final Adjustment**: The colorbar axis should be turned off to match the target visualization.

Based on this analysis, the correct change is to adjust the grid specification and turn off the colorbar axis.

The correct answer is:

D

**Error Analysis:**

GPT-4o over-complicates the problem. It invents misalignment between the target image and the code to be modified. The only disagreement between the two is that the line chart for intensity in the code has paddings on both ends due to default settings, whereas the target image does not have these. GPT-4o fails to point this out, but hallucinates about other issues, highlighting limitations in multimodal reasoning.

**Ground Truth:**

The only misalignment between the code and the target image is that the line chart for intensity in the code has paddings on both ends due to default settings, whereas the target image does not have these. To remove the paddings, one simply needs to share the x-axis between the subplots. **Correct Answer: A**

**Figure 38:** A sample error case in coding.

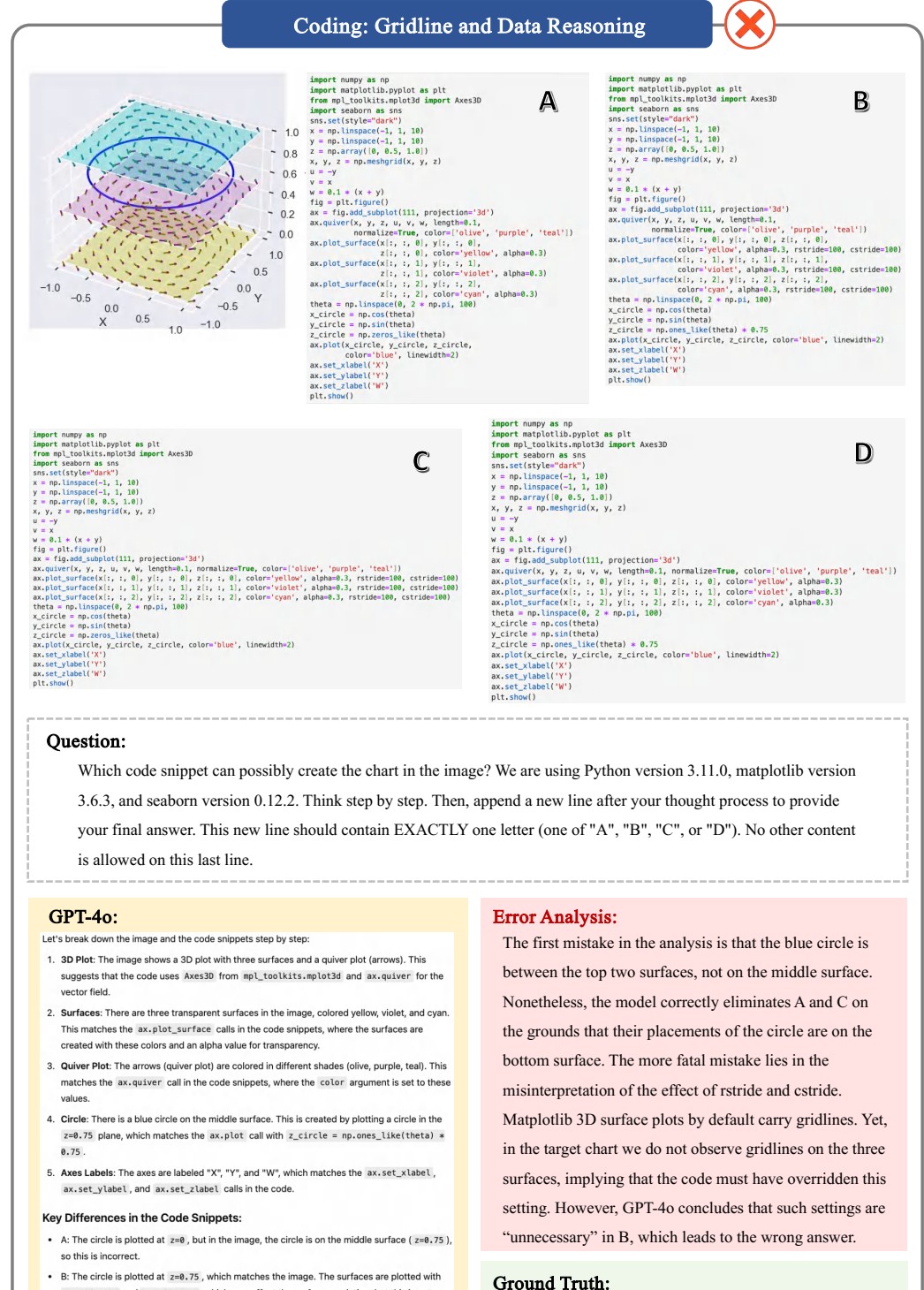

**Figure 39:** A sample error case in coding.

