# OpenReview forum: "Can MLLMs Reason in Multimodality? EMMA: An Enhanced MultiModal ReAsoning Benchmark"
_ICML.cc/2025/Conference — ICML 2025 oral_

### Official Review · Reviewer_Z6QV · 2025-03-11

**Overall Recommendation:** 5

**Summary:**

This paper introduces the EMMA benchmark to evaluate the reasoning capabilities of multimodal LLMs that require the integration of both text and visual cues. The benchmark is curated from existing datasets and supplemented with 1796 newly created questions covering math, chemistry, physics, and coding. A filtering process ensures that questions cannot be answered using text alone. Additionally, the paper evaluates SOTA models, providing a comprehensive analysis of both direct prompting and CoT prompting. The study also explores test-time compute scaling. Results indicate that current SOTA models exhibit a significant performance gap compared to human experts across these evaluations.

**Claims And Evidence:**

Yes

**Essential References Not Discussed:**

There are few works related to MLLMs' spatial reasoning, and all of them provide benchmarks in their papers.

[1] Cheng, An-Chieh, et al. "SpatialRGPT: Grounded Spatial Reasoning in Vision Language Models." arXiv preprint arXiv:2406.01584 (2024).

[2] Liu, Fangyu, Guy Emerson, and Nigel Collier. "Visual spatial reasoning." Transactions of the Association for Computational Linguistics 11 (2023): 635-651.

[3] Nie, Jiahao, et al. "Mmrel: A relation understanding dataset and benchmark in the mllm era." arXiv preprint arXiv:2406.09121 (2024).

**Experimental Designs Or Analyses:**

Yes

**Methods And Evaluation Criteria:**

Yes, only two questions
1) Regarding the filtering mechanisms in L206–L209, what is the motivation for removing questions that could be answered if the original text and generated captions were provided? Captioning an image still relies on visual cues, and if additional reasoning is required, wouldn’t such a question still assess the model’s multimodal reasoning capability?
2) The ‘Code Chose Vis’ setting seems somewhat counterintuitive. It requires directly reading code and mentally visualizing the output, which is quite challenging even for humans, especially for very long and complex code.

**Other Comments Or Suggestions:**

1) It would be helpful to highlight the best and second-best scores in the performance table in the supplementary material, similar to the table in the main paper.
2) Including a breakdown of human performance for EMMA-mini would be valuable as a reference when analyzing the results.

**Other Strengths And Weaknesses:**

The paper is well-written and provides a thorough approach to benchmark creation, comprehensive experiments, and detailed analysis. Reading this paper gave me valuable insights into the current limitations of multimodal reasoning capabilities.

**Questions For Authors:**

1) The benchmarks cover Math, Chemistry, Coding, and Physics, requiring not only multimodal reasoning but also expert knowledge to solve problems. Do you have any plans to expand into more common scenarios that primarily rely on logical reasoning and general knowledge?
2) Regarding the gap between closed-source and open-source models, do you think this is partly due to most open-source MLLMs not incorporating CoT during training? Would this also lead to hallucinations when CoT is enforced during inference?
3) For N=16 Pass@N, the performance is quite high for all three models, with some models even outperforming human experts. Does this suggest that the model already possesses the necessary knowledge and reasoning capability but with low probability to generate? If so, would fine-tuning on similar questions from the benchmark provide a shortcut to boosting performance?

**Relation To Broader Scientific Literature:**

There are many multimodal benchmarks, such as SeedBench, MMMU, RealWorldQA, MuirBench, and VideoMME. Some benchmarks specifically focus on reasoning, such as MATH-V, Visual CoT, and SpatialRGPT. However, some questions in these benchmarks can be answered using only textual cues, or models have already reached saturation on them. This paper introduces several mechanisms to ensure that both text and images are required for reasoning. Additionally, it demonstrates a significant performance gap compared to human experts.

**Theoretical Claims:**

The paper does not have theoretical claims.

---

> ### Author Rebuttal · Authors · 2025-04-01
>
> Thank you for the encouraging and thoughtful review! Below is our response to your questions and suggestions.
>
> **Q1: Why did we filter out questions that can be answered using the text and generated image captions?**
>
> Our enhanced filtering pipeline targets questions requiring deep multimodal reasoning that are difficult to solve through text reasoning or one visual pass. If a question can be solved with text and image captions, it means the necessary visual information can be compressed into text. The question is then solvable with one round of visual perception or shallow visual reasoning, with the rest handled by text reasoning. Instead, we focus on problems requiring back-and-forth textual and visual reasoning, which typically demand repeatedly referring back to the image for multi-hop reasoning or mental simulation. This filtering pipeline allows us to curate problems that assess advanced multimodal reasoning.
>
> **Q2: Counterintuitive task of "Code Choose Vis"**
>
> While "Code Choose Vis" may seem counterintuitive at first, it is tied to how humans typically write visualization code, whereby they envision their desired chart, code it, and mentally check if the code achieves their intention. The critical step of asking oneself, "is the code doing what I want it to do?" is precisely what "Code Choose Vis" problems target. This ability is also prerequisite to debugging visualizations, where one must understand what the current code produces before making corrections.
>
> **Q3: Plans to expand EMMA to more general domains**
>
> While most questions in EMMA require domain knowledge, many math questions do not. For example, simulating how a 2D shape will look after transformations or finding patterns in a sequence of shapes relies primarily on logic and general knowledge rather than math expertise.
>
> In scoping for EMMA, we explored various disciplines for multimodal reasoning questions meeting our strict criteria. We found that most such problems are typically considered "logic" or "spatial reasoning" tasks, which we ultimately categorized under math. In other domains, such problems are much harder to source or create. Nonetheless, we are very interested in incorporating more general domain multimodal reasoning questions and are actively exploring strategies for expansion. Thank you for this great suggestion!
>
> **Q4: Why does CoT prompting not help with open-source models?**
>
> We observed that CoT prompting generally improves performance for closed-source models, but tends to hurt performance for open-source models. One possible reason is that open-source models do not effectively leverage language to assist in multimodal reasoning tasks where language could be beneficial. For example, language can often help grounding in multi-hop counting, so it should be theoretically possible to improve performance on this task with CoT, but open-source models fail to capitalize on this.
>
> Previous work has shown that the quality of training data is crucial for CoT effectiveness. For example, [1] has found that filtering out anomalous samples (e.g., repetitive patterns) can improve CoT performance. Your point regarding the lack of CoT supervision during training aligns with this perspective: incorporating high-quality CoT data can indeed be seen as one way to improve the quality of training data for MLLMs.
>
> Without access to the training data and pipelines of the tested models, it is difficult to pinpoint the reason behind this divergence. Nonetheless, we believe that multiple factors might contribute to this phenomenon, and a more systematic understanding of CoT prompting in multimodal settings remains an open and important direction for future research.
>
> [1] https://arxiv.org/pdf/2412.05271
>
> **Q5: Does high Pass@16 indicate latent capability and can fine-tuning on similar data boost performance?**
>
> That is a keen observation. We agree that strong Pass@16 performance may suggest latent reasoning capabilities. However, as you noted, even if models possess relevant knowledge, they have a low probability of applying it correctly. Moreover, correct answers can sometimes stem from flawed reasoning, especially for multiple-choice questions.
>
> The idea of finetuning on similar questions can indeed be promising. Prior work [2] has highlighted the nuanced interplay between memorization and generalization in finetuning LLMs on logical reasoning, and it would be interesting to investigate this further in the multimodal setting—an interesting direction for future work.
>
> [2] https://arxiv.org/pdf/2410.23123
>
>
> > Suggestion 1: Discussing additional relevant work
>
> Thank you for these highly valuable and relevant references. We will include and discuss them in an updated version.
>
> > Suggestion 2: Highlighting top scores in tables and including human performance breakdown on EMMA-mini
>
> Thank you for the thoughtful suggestion. We will make sure to more clearly highlight results and include a category-level breakdown of human performance.

---

### Official Review · Reviewer_8w8m · 2025-03-13

**Overall Recommendation:** 4

**Summary:**

This paper introduces EMMA, a visual question answering benchmark requiring multimodal reasoning. EMMA includes questions in four domains: math, physics, chemistry, and coding. The questions in EMMA are filtered so that they are not answerable based on only the image captions and questions. The experiments show that both open-source and closed-source MLLMs fall significantly short of human expert performance. Besides, the effect of chain-of-thoughts prompting and test-time compute scaling on the evaluated models are discussed.

**Claims And Evidence:**

1. The paper claims that the state-of-the-art MLLMs struggle with multimodal reasoning. This is supported by the experimental results that the MLLMs achieve much lower accuracy than human experts on the proposed EMMA benchmark.
2. The paper claims that the EMMA benchmark cannot be addressed by reasoning on solely one modality. The data curation process uses this condition to filter questions with GPT-4o as the reasoning model. However, other evaluted models are not tested in the single-modality setting to support this claim.

**Essential References Not Discussed:**

N/A

**Experimental Designs Or Analyses:**

I checked the soundness/validity of the experimental designs and analysis and did not find any issue.

**Methods And Evaluation Criteria:**

In the experiments, the MLLMs are not only evaluated in direct prompting, but also with chain-of-thoughts prompting and test-time compute scaling. A group of commonly used approach to boost reasoning performance are used to demonstrate the weak performance on the proposed benchmark.

**Other Comments Or Suggestions:**

While the domain of the benchmark is limited, I highly appreciate the level of challenge of the benchmark and the thorough evaluation of MLLMs and reasoning techniques. So, I give a score of 3 as my initial recommendation.

**Other Strengths And Weaknesses:**

Other Strengths:
1. The proposed benchmark is well presented. The paper shows detailed statistics and example questions in the benchmark.
2. 1,796 questions are newly collected by domain experts, which are valuable to the community.
3. A few advanced techniques to improve MLLM reasoning are evaluated and analyzed in the benchmark. This demonstrates the advantages and limitations of these techniques.
4. The paper is well-written and easy to follow. The proposed benchmark is well presented.

Other Weaknesses:
1. Although the data curation filters out the questions that can be answered based on questions and image captions, only GPT-4o, Llama-3, and Qwen2 are used in this process. So, the weak questions filtered out are restricted to the capability of these models. It would be more convincing if experiments can be conducted to evaluate other MLLMs in the "question only" and "caption and question" settings.
2. While EMMA includes challenging multimodal reasoning questions, they are restricted to the domains of math, physics, chemistry, and coding. This would lead to two issues:

+ Answering the questions in EMMA requires strong domain knowledge. Therefore, the accuracy on the benchmark cannot directly reflect the reasoning capability of a model. The lack of domain knowledge can also cause weak performance.

+ The domain of the benchmark is limited. It does not include more diverse domains, e.g., geography and biology. More importantly, none of the images involved in the benchmark are realistic images. While it is acceptable to propose a benchmark in specific domains, the paper should discuss it as a limitation. Besides, the name "Enhanced Multimodal Reasoning Benchmark" exaggerates the contribution of the benchmark because it is domain-specific instead of a general-domain benchmark.

**Questions For Authors:**

1. In L 193-194, what does "a single visual pass" mean?
2. What does "Pass@N" mean in Table 3?

These should be clarified in paper writing.

**Relation To Broader Scientific Literature:**

The previous multimodal reasoning benchmarks have shortcuts in the questions so that they may fail to evaluate the multimodal reasoning capability. The newly proposed EMMA benchmark filters the questions in previous benchmarks and also collects new questions. The questions in EMMA are ensured that they are not answerable based off the questions and image captions, making is a challenging benchmark for multimodal reasoning evaluation.

**Theoretical Claims:**

This paper does not involve theoretical claims.

---

> ### Author Rebuttal · Authors · 2025-04-01
>
> Thank you for your detailed review and great questions! We hope our response below helps address them.
>
> **Concern 1: Only 3 models are used in filtering. Other models might be able to solve the retained questions in the text-only setting**
>
> (1) The 3 models used were among the strongest available at the time. If none could consistently solve a question with text and captions, other models are unlikely to succeed under the same conditions.
>
> (2) We also wish to emphasize that MLLM-based filtering is **only the first step in our data pipeline** (Section 3.2). This was followed by manual filtering to ensure retained questions genuinely required multimodal reasoning. In fact, 16.4% of questions were removed at this stage because they did not involve substantial multimodal reasoning.
>
> (3) Following your suggestion, we tested more recent MLLMs in the "caption and question" setting. We focused on math to test whether our filter separates visual-reasoning-heavy questions from light ones (hereafter "heavy" and "light" questions).
> We selected:
> - **100 "heavy"** math questions from EMMA-mini
> - **100 "light"** math questions randomly sampled from the filtered-out pool
>
> We evaluated 5 SOTA MLLMs using only the question and caption, and also report a *full-input* setting (text + image) on heavy questions:
> |Model|Visual Reasoning Light (Text + Caption)|Visual Reasoning Heavy (Text + Caption)|Visual Reasoning Heavy (Text + Image)|
> |---------------------------|------------------------------------|--------------------------------------|-------------------------------------|
> |Claude 3.7 Sonnet|74|41|45|
> |Gemini 2.0 Flash Thinking|72|31|34|
> |GPT-4o|76|31|27|
> |Qwen2.5-VL-72B|74|27|39|
> |InternVL2.5-78B|56|23|31|
>
> - In the caption-only setting, all models perform significantly better on light questions than on heavy ones, indicating that the filtered-out questions can largely be solved via text-based reasoning and that our filtering pipeline effectively identifies visual-reasoning-heavy questions;
> - On heavy questions, providing the image offers only marginal gains over captions. Even advanced MLLMs still struggle to utilize visual information for reasoning, reinforcing the motivation for EMMA.
>
> **Concern 2: Lack of domain knowledge can cause weak performance**
>
> We agree that a lack of domain knowledge can lead to weak performance on EMMA. However, we believe that the primary factor behind the underperformance of SOTA models on EMMA is not the lack of domain knowledge,  but poor multimodal reasoning skills. Recent benchmarks suggest that SOTA models are equipped with strong domain knowledge. For example, models have achieved 87.7\% accuracy on GPQA Diamond (which tests graduate-level science knowledge), 78.2\% on MMMU (which spans many subjects), and 86.5\% on AIME (a high school-level math competition). **This suggests that the knowledge required for EMMA is largely present in current models.** Hence, we believe that poor multimodal reasoning skills are the bottleneck.
>
> **Concern 3: EMMA is limited to four domains and does not include natural images**
>
> In scoping EMMA, we explored many disciplines for questions meeting our strict criteria. We found that in many domains the primary challenge lies in knowledge rather than in multimodal reasoning. For example, many biology questions involve tasks such as labeling parts of a complex diagram. These tasks often hinge more on knowledge than on reasoning over multimodal information. Constructing/curating multimodal reasoning questions in these domains is also more difficult. That said, we appreciate your suggestion and plan to involve more disciplines in future work, such as law and medicine.
>
> We also agree that including natural images would be valuable. However, sourcing high-quality, reasoning-focused problems using natural images has proven to be challenging. Benchmarks like MMMU also include very few natural images. In math, physics, and chemistry, most problems are accompanied by simplified diagrams or sketches designed to reduce ambiguity and clarify assumptions. We will explore ways to design or curate problems involving natural imagery in future work.
>
> > Q1: Meaning of "a single visual pass"
>
> We operationalize "a single visual pass" as generating image captions with GPT-4o. Multimodal reasoning questions often require looking at an image multiple times for multi-step/multi-hop reasoning. If a question can be solved with text and captions, its visual content can likely be compressed into text with shallow visual perception/reasoning, and the rest can be handled by text reasoning. Whereas previous work, BLINK [1], assesses perception beyond recognition, we target images that are necessary for reasoning beyond perception.
>
> [1] https://arxiv.org/abs/2404.12390
>
> > Q2: Meaning of Pass@N
>
> Pass@N refers to the accuracy when generating N responses per question and checking if **any** of them is correct. Thus, it is an upper bound for accuracy from test-time scaling.

---

> > ### Comment · Reviewer_8w8m · 2025-04-04
> >
> > Thank the authors for the responses. They addressed my concerns and I have raised my rating to 4 (accept).

---

### Official Review · Reviewer_PLqU · 2025-03-14

**Overall Recommendation:** 4

**Summary:**

This paper introduces EMMA, a novel benchmark designed to evaluate the vision-language reasoning capabilities of MLLMs.
Unlike existing benchmarks that focus on shallow visual understanding or text-dominated problem-solving, EMMA emphasizes tasks where solutions inherently require iterative interaction between visual and textual reasoning.
The benchmark covers four domains — mathematics, physics, chemistry, and coding — and presents challenges such as 3D spatial transformations, chemical structure analysis, and multi-step simulation. It consists of 992 filtered questions from existing datasets and 1,796 newly curated questions developed in collaboration with domain experts.
Evaluation of SOTA MLLMs reveals critical limitations: techniques like chain-of-thought prompting and test-time computation scaling (e.g., majority voting) provide only marginal improvements. Moreover, the models struggle with tasks requiring precise spatial simulation or the ability to leverage visual aids for enhanced efficiency, exposing difficulties in fine-grained spatial reasoning, multi-hop visual-text integration, and generating effective visual reasoning steps.
The authors argue that current architectures and training paradigms are inadequate for supporting deep multimodal reasoning, calling for innovations to better integrate different modalities.

**Claims And Evidence:**

Yes, the claims made in the submission are supported by clear and convincing evidence.

**Essential References Not Discussed:**

Yes. the related works are sufficient.

**Experimental Designs Or Analyses:**

Yes, the soundness and validity of the experimental designs and analyses were carefully reviewed. No significant issues were identified.

**Methods And Evaluation Criteria:**

Yes

**Other Comments Or Suggestions:**

No

**Other Strengths And Weaknesses:**

[Strengths]

- 1) This paper is rich and well-grounded, providing detailed definitions of fine-grained labels, a clear data construction process, and comprehensive experimental results on the key issues currently being explored by the research community.

- 2) The authors have contributed a large number of novel multimodal disciplinary questions through manual collection and expert annotation.

[Weaknesses]

- 1) The term "organically reason" mentioned in the abstract is somewhat confusing. How should the term "organic" be interpreted in this context?

- 2) Using the criterion of "whether an image caption can replace the visual input" to filter the data intuitively seems to only exclude questions that are "difficult to describe in words," but it may not necessarily ensure that the image content is "more aligned with visual perception than with visual reasoning." For example, identifying a symbol in a musical staff or counting the zeros of a function graph whose expression cannot be explicitly determined might be incorrectly sampled. I wonder if this type of bias exists, providing some error analysis could make the work more robust.

**Questions For Authors:**

Pleaser refer to Part of [Other Strengths And Weaknesses]

**Relation To Broader Scientific Literature:**

N/A

**Theoretical Claims:**

This work is a benchmark study and does not involve theoretical derivations.

---

> ### Author Rebuttal · Authors · 2025-04-01
>
> Thank you for your encouraging review and insightful questions. We provide our responses below.
>
> **Q1: What does "organically reason" mean?**
>
> By "organically reason over and with both text and images", we refer to the integrated way humans seamlessly blend visual and textual information during reasoning processes. To measure advanced multimodal reasoning capabilities, we target questions that require both modalities working together, where neither text nor images alone are sufficient. Models must dynamically engage with both reasoning channels and effectively fuse textual and visual information to succeed on our benchmark.
>
> **Q2: The filtering pipeline seems to only retain problems that are difficult to describe in words, which can include questions that are more difficult in visual perception than in visual reasoning.**
>
>
> We agree that our first-step filtering, which removes questions solvable by models given only question text and image captions, may retain examples that are difficult to describe in words, but do not necessarily require strong visual reasoning. For instance, identifying a symbol in a musical staff might be retained simply because the symbol is visually subtle or hard to express textually, even though solving the task does not demand deep visual reasoning.
>
> To address this potential bias, we installed a second filtering step. Specifically, we manually reviewed the remaining set and constructed a taxonomy that emphasizes multimodal reasoning skills. For example, for math, we identified categories such as 3D spatial simulation and pattern inference. **We then used GPT-4o to categorize the questions according to this taxonomy, followed by a final round of manual verification to ensure quality and relevance.** This process helped ensure that the retained questions require visual reasoning.
>
> Here we provide an [example](https://huggingface.co/datasets/MathLLMs/MathVision/viewer/default/test?views%5B%5D=test&row=3) from the MathVision dataset that is retained after the first-stage filtering but removed during manual verification. Solving the problem requires correctly perceiving all the digits in the image, which MLLMs struggle with. For instance, GPT-4o recognizes the digits at the bottom as "2" instead of "3", resulting in an incorrect answer. Although the question cleared the first-stage filter, we excluded it as it primarily tests visual perception rather than visual reasoning.
>
> To further support our claim, we provide the following statistics from the MathVision dataset: out of 3,040 total questions, 2,195 were retained after the initial model-based filtering. **However, only 668 questions (30.43\%) from that set were ultimately included in EMMA, based on our taxonomy and human verification.** This highlights that our filtering method applies a stricter standard to ensure a focus on multimodal reasoning. As noted in our error analysis (see Figure 5), MLLMs sometimes still fail on EMMA questions due to perceptual errors. However, we do not view these errors as contradictory to the multimodal-reasoning-based nature of the tasks.  Rather, they may reflect perceptual limitations that are prerequisites for successful multimodal reasoning—an important challenge in its own right. In other words, while the immediate failure may stem from perception, the questions still require multimodal reasoning to be solved, highlighting a compound challenge that current models have yet to overcome.
>
> In summary, we have taken careful steps to ensure that the problems included in EMMA emphasize visual reasoning over low-level visual perception. We appreciate your thoughtful comment, as it raises an important distinction and gives us the opportunity to clarify our data curation process more thoroughly.

---

> > ### Comment · Reviewer_PLqU · 2025-04-02
> >
> > Thanks for the authors' response, which has addressed my concern.

---

### Official Review · Reviewer_QCsL · 2025-03-23

**Overall Recommendation:** 4

**Summary:**

This paper proposed a benchmark EMMA (Enhanced MultiModal reAsoning) to feature questions that are difficult to solve by relying solely on text-based reasoning or a single visual pass, covering math, physics, chemistry, and coding domains with 2,788 questions. Ten state-of-the-art MLLMs are further evaluated on the benchmark, revealing (1) a substantial performance gap compared to human experts and (2)  techniques such as Chain-of-Thought prompting and test-time compute scaling offering only marginal gains.

**Claims And Evidence:**

Yes

**Essential References Not Discussed:**

No

**Experimental Designs Or Analyses:**

Yes

**Methods And Evaluation Criteria:**

Yes

**Other Comments Or Suggestions:**

Please see weaknesses

**Other Strengths And Weaknesses:**

Strengths:

* The proposed benchmark reveals the gap between SOTA MLLMs with human experts, and will serve as a strong benchmark on multimodal reasoning evaluation.

* The experiments reveal that test-time scaling strategies cannot achieve a strong performance yet, calling for new reasoning strategies in the future.

* The experiments are comprehensive, covering many SOTA MLLMs. The curated data quality is also high.

Weaknesses:

* The paper is a pure benchmark without a new method. This is not a reason for rejection, but is a weakness or limitation on technical contributions.

**Questions For Authors:**

Please see weaknesses

**Relation To Broader Scientific Literature:**

No

**Theoretical Claims:**

The paper does not include theoretical proofs.

---

> ### Author Rebuttal · Authors · 2025-04-01
>
> Thank you very much for your thoughtful and careful reading!
>
> As you have pointed out, our benchmark provides a test suite that reveals significant limitations of even the most advanced MLLMs in handling complex multimodal reasoning tasks. Although state-of-the-art MLLMs have recently achieved strong results on many challenging benchmarks, their poor performance on our benchmark highlights key gaps in their reasoning capabilities. **We attribute this, in part, to our enhanced filtering pipeline, which allowed us to surface questions that genuinely require multimodal reasoning.** Importantly, this pipeline is reusable and could serve as a framework for constructing future benchmarks aimed at other aspects of multimodal understanding.
>
> In addition, we utilized our benchmark as a testbed to provide technical evaluation and insights into advanced techniques for enhancing MLLM reasoning. By comparing model performance with and without CoT, and evaluating different scaling approaches—such as majority voting, best-of-N, and tournament selection—we highlight the limitations of current prompting and inference-time methods for complex visual reasoning. These technical insights also would not have been possible without a carefully curated benchmark.
>
> By exposing these limitations, our work points to broader issues, such as potential shortcomings in current model architectures or a mismatch between existing training paradigms and the demands of complex multimodal reasoning. We hope this will encourage further research into more robust multimodal architectures and training strategies.
>
> We deeply appreciate your constructive suggestions. Following your advice, we plan to explore new methods to tackle these challenges in future work, and we remain committed to advancing the evaluation and development of MLLMs’ visual reasoning capabilities.
>
> Thank you again for your insightful and encouraging feedback!

---

### Decision · Program_Chairs · 2025-05-01

**Decision:**

Accept (oral)

**Comment:**

The paper introduces EMMA, a high-quality benchmark for evaluating the multimodal reasoning abilities of MLLMs across domains such as math, physics, chemistry, and coding. Reviewers unanimously appreciated the benchmark’s novelty, rigorous filtering pipeline, and insightful evaluations, which expose current models’ limitations in visual-textual integration. While some initial concerns were raised about the domain scope and methodology, the authors provided thorough and convincing responses that addressed reviewer queries, leading to rating increase. Given the paper’s strong motivation, comprehensive evaluation, and community value, all reviewers now recommend acceptance. AC agrees that this is a solid contribution and should be accepted.